# Over-Dose Lithium Toxicity as an Occlusive-like Syndrome in Rats and Gastric Pentadecapeptide BPC 157

**DOI:** 10.3390/biomedicines9111506

**Published:** 2021-10-20

**Authors:** Sanja Strbe, Slaven Gojkovic, Ivan Krezic, Helena Zizek, Hrvoje Vranes, Ivan Barisic, Dean Strinic, Tatjana Orct, Jaksa Vukojevic, Spomenko Ilic, Eva Lovric, Darija Muzinic, Danijela Kolenc, Igor Filipčić, Zoran Zoricic, Darko Marcinko, Alenka Boban Blagaic, Anita Skrtic, Sven Seiwerth, Predrag Sikiric

**Affiliations:** 1Department of Pharmacology, School of Medicine, University of Zagreb, 10000 Zagreb, Croatia; strbes@gmail.com (S.S.); slaven.gojkovic.007@gmail.com (S.G.); ivankrezic94@gmail.com (I.K.); zizekhelena@gmail.com (H.Z.); hrvoje.vranes@gmail.com (H.V.); inbarisic@gmail.com (I.B.); destrinic@gmail.com (D.S.); jaksavukojevic@gmail.com (J.V.); silic67@gmail.com (S.I.); igor.filipcic@pbsvi.hr (I.F.); zoran.zoricic@kbcsm.hr (Z.Z.); predstojnik.psi@kbc-zagreb.hr (D.M.); abblagaic@mef.hr (A.B.B.); 2Institute for Medical Research and Occupational Health, 10000 Zagreb, Croatia; torct@imi.hr (T.O.); darija.muzinic@gmail.com (D.M.); 3Department of Pathology, School of Medicine, University of Zagreb, 10000 Zagreb, Croatia; eva.lovric@kb-merkur.hr (E.L.); danijela.kolenc@mf.uni-lj.si (D.K.); sven.seiwerth@mef.hr (S.S.)

**Keywords:** lithium toxicity, occlusive-like syndrome, rats, gastric pentadecapeptide BPC 157

## Abstract

Due to endothelial impairment, high-dose lithium may produce an occlusive-like syndrome, comparable to permanent occlusion of major vessel-induced syndromes in rats; intracranial, portal, and caval hypertension, and aortal hypotension; multi-organ dysfunction syndrome; brain, heart, lung, liver, kidney, and gastrointestinal lesions; arterial and venous thrombosis; and tissue oxidative stress. Stable gastric pentadecapeptide BPC 157 may be a means of therapy via activating loops (bypassing vessel occlusion) and counteracting major occlusion syndromes. Recently, BPC 157 counteracted the lithium sulfate regimen in rats (500 mg/kg/day, ip, for 3 days, with assessment at 210 min after each administration of lithium) and its severe syndrome (muscular weakness and prostration, reduced muscle fibers, myocardial infarction, and edema of various brain areas). Subsequently, BPC 157 also counteracted the lithium-induced occlusive-like syndrome; rapidly counteracted brain swelling and intracranial (superior sagittal sinus) hypertension, portal hypertension, and aortal hypotension, which otherwise would persist; counteracted vessel failure; abrogated congestion of the inferior caval and superior mesenteric veins; reversed azygos vein failure; and mitigated thrombosis (superior mesenteric vein and artery), congestion of the stomach, and major hemorrhagic lesions. Both regimens of BPC 157 administration also counteracted the previously described muscular weakness and prostration (as shown in microscopic and ECG recordings), myocardial congestion and infarction, in addition to edema and lesions in various brain areas; marked dilatation and central venous congestion in the liver; large areas of congestion and hemorrhage in the lung; and degeneration of proximal and distal tubules with cytoplasmic vacuolization in the kidney, attenuating oxidative stress. Thus, BPC 157 therapy overwhelmed high-dose lithium intoxication in rats.

## 1. Introduction

We hypothesized that the higher therapeutic and supratherapeutic lithium levels known to impair endothelium-dependent blood vessel relaxation, as an additional mechanism contributing to lithium toxicity (e.g., renal and neurological) [1], may be related to a worse syndrome of vascular impairment—an “occlusion-like” syndrome [2,3,4,5,6,7,8,9]—and thereby extensive lithium toxicity, peripherally and centrally.

Recently, we demonstrated that the stable gastric pentadecapeptide BPC 157 (for a review see [10,11,12,13,14]) may counteract the adverse effects of the administration of a huge dose of lithium sulfate (500 mg/kg ip) in rats [15]; dose selection in accordance to the counteracted magnesium-high dose toxicity [16] showed that many direct targets of lithium are likely magnesium-dependent intracellular enzymes [17]. Otherwise, an immediate severe syndrome was produced (severe muscular weakness and prostration, reduced muscle fibers, myocardial infarction, and edema of various brain areas (with the most prominent being in the cerebral cortex)) [15]. This deterioration, which appeared with subsequent applications of lithium, was also counteracted by concomitant application of BPC 157 [15]. These findings [15] were suggestive of additional brain, heart, lung, liver, kidney, and gastrointestinal tract lesions, which may, due to the lithium-induced vascular impairment [1], produce an additional high-dose lithium syndrome, i.e., an occlusion-like syndrome, with intracranial, portal and caval hypertension; aortal hypotension; and generalized peripheral and central thromboses. To date, the possibility that lithium toxicity consequently appears as an occlusion-like syndrome [2,3,4,5,6,7,8,9] has not been investigated or appropriately combined with lithium toxicity. The multi-organ dysfunction syndrome due to lithium-induced vascular impairment [1] may be akin to those syndromes induced in rats with occluded major vessels [2,3,4,5,6,7,8], peripherally [2,3,4,5,6,7] and centrally [8]. Because BPC 157 therapy consistently counteracted all of these occlusive syndromes [2,3,4,5,6,7,8], it may be effective in rats with high-dose lithium intoxication and may largely counteract the lithium syndrome. In support of this idea, a multi-organ dysfunction syndrome characterized by lithium-induced vascular impairment [1], alcohol-induced vascular impairment [9], and absolute alcohol instillation in the rat stomach, produced a similar occlusion-like syndrome of peripheral and central multi-organ dysfunction; intracranial, portal, and caval hypertension; aortal hypotension; and generalized peripheral and central thromboses. BPC 157 therapy fully counteracted these effects [9].

Furthermore, this may be relevant to the reports of thrombosis, and thereby blood stasis, in patients treated with lithium [18,19,20]. For example, Budd–Chiari syndrome (suprahepatic occlusion of the inferior caval vein) was attenuated/eliminated with BPC 157 therapy [10], including the counteraction of the fatal outcome of the syndrome [5]. Heart dysfunction, lung lesions (i.e., time-dependent and time-independent features resembling the exudative phase features of acute respiratory distress syndrome (ARDS)), liver failure, gastrointestinal lesions, widespread arterial and venous thromboses, severe portal and caval hypertension, and aortal hypotension were all counteracted [5]. A similar syndrome, including intracranial hypertension, was noted with the occlusion of the superior sagittal sinus [8], and the effects of BPC 157 therapy (in addition to counteracting the intracranial (superior sagittal sinus) hypertension, brain swelling, and lesions) were similar to those seen in previous peripheral vessel occlusion studies [2,3,4,5,6,7]. The activation of the collateral loops occurred in a particular manner (i.e., left superior caval vein-azygos vein-inferior caval vein shunt (Budd–Chiari syndrome) [5] and recruited (para)sagittal venous collateral circulation (central venous occlusion)) [8]. Thus, BPC 157 activated a “bypassing key” (i.e., collateral pathways reliant on the injurious occlusion) [2,3,4,5,6,7,8], suggesting the beneficial effects commonly noted in other vascular injury studies [21,22,23,24,25,26], which was in essential competition with the commonly present Virchow’s triad (endothelium lesion, hypercoagulability, and stasis) and its resolution. In addition, BPC 157 interacts with several molecular pathways [27,28,29,30,31,32,33,34,35]. Similarly, considering the NO system-related endothelial impairment induced by lithium [1], we emphasized its modulatory effects on the NO [36] and prostaglandin systems [37], the vasomotor tone, and the activation of the Src-caveolin-1-eNOS pathway [29]. Moreover, the thrombocyte function was maintained (without interfering with coagulation) [38], in addition to their actions as stabilizers of cellular junctions [27] and free radical scavengers [39,40,41], particularly in vascular occlusion studies [2,3,4,5,6,7,21,22,23,24].

Lithium has been used for over half a century to treat affective disorders. Due to lithium’s narrow therapy window [17], BPC 157 has been implicated in the functioning of the brain–gut axis and has been shown to interact, as a modulator, with several neurotransmitters (for a review, see, i.e., [12,14]). It may counteract not only the acute and chronic toxicities of amphetamines [42,43,44], but also neuroleptic-induced catalepsy [44,45], in addition to other neuroleptic adverse effects (i.e., somatosensory disorientation [45], gastric ulcers [45,46,47], sphincter dysfunction [41], and prolonged QT-interval [48]) and L-NAME- and NOS blocker-induced catalepsy [44]. BPC 157 has a particular antidepressant effect [49], may counteract serotonin syndrome and induce serotonin release in particular brain areas (i.e., nigrostriatal areas) [50], and interact with serotonin in the gastrointestinal tract [35]. As was proposed for lithium [51], BPC 157 also acts by blocking the development of supersensitive DA receptors [42]. Recently, BPC 157 was found to counteract schizophrenia-like positive symptom models through a NO system-related effect [44]. Moreover, BPC 157 counteracted the effects of potassium or magnesium overdose [16,52,53].

Thus, following the initial report of high-dose lithium-induced endothelial impairment [1], this may be the first study to investigate a high-dose lithium regimen leading to a full occlusive syndrome (peripherally and centrally), i.e., a multi-organ dysfunction syndrome similar to those described previously in rats, and characterized by occlusion of major vessels [2,3,4,5,6,7,8], permanent intracranial (superior sagittal sinus) hypertension, portal hypertension, caval hypertension, and aortal hypotension. Of note, instead of permanent irremovable occlusion of one or two major vessels [2,3,4,5,6,7,8], as a prime target, the high-dose lithium regimen induced endothelial impairment and may have multiple targets. However, it is nonetheless intended that the therapy can be used to elucidate the activated rescuing bypassing collateral pathway as a key to the recovery of lithium toxicity as an occlusive-like syndrome. The activated rescuing bypassing collateral pathway (i.e., the azygos vein, and the inferior caval vein–azygos vein–superior caval vein pathway) was also involved in the recovery of the intragastric absolute alcohol-induced occlusion-like syndrome [9]. We hypothesized that, in accordance with its previous therapeutic effects in occlusive syndromes [2,3,4,5,6,7,8] and alcohol-induced occlusion-like syndrome [9], BPC 157 administration would counteract both peripheral (portal hypertension, caval hypertension, and aortal hypotension) and central disturbances (i.e., brain swelling/lesions and intracranial hypertension), in addition to organ lesions and oxidative stress, in rats treated with a high-dose lithium therapy. The serum concentration of lithium was assessed at a suitable time to exclude the possibility that the relatively good results in the treatment group may be due to the lower drug concentration in the blood.

## 2. Materials and Methods

### 2.1. Animals

This study was conducted with 12 week old, male Albino Wistar rats having a body weight of 200 g, which were randomly assigned to six rats/group/interval. Rats were bred in-house at the Pharmacology Animal Facility, School of Medicine, Zagreb, Croatia. The animal facility was registered by the Directorate of Veterinary (Reg. No: HR-POK-007). Laboratory rats were acclimated for 5 days and randomly assigned to their respective treatment groups. Laboratory animals were housed in polycarbonate (PC) cages under conventional laboratory conditions at 20–24 °C, relative humidity of 40–70%, and noise level of 60 dB. Each cage was identified with dates, number of the study, group, dose, number, and sex of each animal. Fluorescent lighting provided illumination for 12 h per day. A standard Good Laboratory Practice (GLP) diet and fresh water were provided ad libitum. Animal care was in compliance with the standard operating procedures (SOPs) of the animal facility and the European Convention for the Protection of Vertebrate Animals used for Experimental and other Scientific Purposes (ETS 123). This study was approved (Number: 641-01/17-02101; Date: 02 November 2017 (by the local ethics committee. Ethical principles of the study complied with the European Directive 010/63/E, the Law on Amendments to the Animal Protection Act (Official Gazette 37/13), the Animal Protection Act (Official Gazette 135/06), the ordinance on the protection of animals used for scientific purposes (Official Gazette 55/13), recommendations of the Federation of European Laboratory Animal Science Associations (FELASA), and the recommendations of the Ethics Committee of the School of Medicine, University of Zagreb. The experiments were assessed by observers blinded to the treatment.

### 2.2. Drugs

Stable gastric pentadecapeptide BPC 157, a partial sequence of the human gastric juice protein BPC freely soluble in water at pH 7.0 and in saline, was administered as described previously [2,3,4,5,6,7,8,9], without the use of a carrier or peptidase inhibitor. BPC 157 (GEPPPGKPADDAGLV, molecular weight 1419; Diagen, Ljubljana, Slovenia) was prepared as a peptide with 99% high-performance liquid chromatography (HPLC) purity, with 1-des-Gly peptide being the main impurity. The dose and application regimens were as described previously [2,3,4,5,6,7,8,9]. Lithium sulfate (Sigma, St. Louis, MO, USA) was used.

### 2.3. Experimental Protocol

Using the protocol described before [16], 24 h before the experiments, the rats were exercised by 20 consecutive front and hind paw grips on a vertical grid, gently held at the base of their tail as previously described [54].

The same procedure was repeated on 3 consecutive days. Immediately after administration of lithium sulfate (500 mg/kg ip), rats received 10 µg/kg BPC 157, 10 ng/kg BPC 157, or 5 mL/kg saline intraperitoneally, and after 20 min, the rats were placed on an upside-down grid. If the rats fell, they were continuously placed again within 1 min until the end of the 8 min period. At 3 h after the end of each session, the rats were euthanized.

Recording of the brain swelling in rats was performed 15 min after a complete calvariectomy. Briefly, six burr holes were drilled in three horizontal lines, all of them medial to the superior temporal lines and temporalis muscle attachments. The two rostral burr holes were placed just basal from the posterior interocular line, the two basal burr holes were placed just rostral to the lambdoid suture (and transverse sinuses) on both sides, and the middle two burr holes were placed in the line between the basal and rostral burr holes. The procedure was undertaken 3 h after each of the sessions. Alternatively, lithium administration was recorded as time 0, and brain presentation was recorded in healthy rats 15 min before lithium administration (−15 min → 0). Then, lithium 500 mg/kg in saline was given intraperitoneally (time 0), and the 0 → +3 min period was recorded. Thereafter, the +3 min (5 mL/kg saline intraperitoneally) → +6 min period was recorded. Then, the administration of BPC 157 was recorded during the next 3 min period (+6 min → +9 min period).

A laparotomy was performed for the corresponding presentation of the peripheral veins (superior mesenteric, inferior caval, and azygos veins), and a camera attached to a VMS-004 Discovery Deluxe USB microscope (Veho, Dayton, OH, USA) was used for recording. The procedure was undertaken 3 h after each of the sessions.

*Muscular weakness.* The amount of time the rats could hold on to the grid reflected the grade of fatigue and muscle weakness. As described previously, the scoring assessment (0–5, healthy rats presented with scores of 4 and 5) was carried out in 1 min intervals until the end of the session as follows: 0/5: immediately falling, no contraction, hunched posture with flaccid paralysis; 1/5: falling (\5 s), muscle flicker, but no movement, hunched posture upon falling; 2/5: falling (\10 s), movement possible, but not against gravity, hunched posture upon falling; 3/5: falling (\20 s), movement possible against gravity, but not against resistance by the examiner, hunched posture upon falling; 4/5: no obvious fatigue, movement possible against some resistance by the examiner, normal posture and activity upon falling (150 s); 5/5: no fatigue, movement possible against significant resistance by the examiner, normal posture and activity upon falling (150 s).

*ECG recording.* ECGs were recorded continuously in deeply anesthetized rats for all three main leads by positioning stainless steel electrodes on all four limbs using an ECG monitor with a 2090 programmer (Medtronic, Minneapolis, MN, USA) connected to a Waverunner LT342 digital oscilloscope (LeCroy, Chestnut Ridge, NY, USA) prior to sacrifice. This arrangement enabled precise recordings, measurements, and analyses of ECG parameters [2,3,4,5,6,7,8,9]. The procedure was undertaken at 3 h after each of the sessions.

Thrombus assessment. On being euthanized, the superior mesenteric vein and superior mesenteric artery were removed from the rats, and the clots were weighed [2,3,4,5,6,7,8,9].

Superior sagittal sinus, portal and inferior caval veins, and abdominal aorta pressure recordings. As described previously [2,3,4,5,6,7,8,9], recordings were made in deeply anesthetized rats with a cannula (BD Neoflon™ Cannula, BD Switzerland, Eysins, Switzerland) connected to a pressure transducer (78534C MONITOR/TERMINAL; Hewlett Packard, Palo Alto, CA, USA) that was inserted into the superior sagittal sinus, portal and inferior caval veins, and abdominal aorta at the level of the bifurcation at 3 h after each session, after 5 min of recording. For the superior sagittal sinus pressure recordings, we made a single burr hole in the rostral part of the sagittal suture just above the superior sagittal sinus and cannulated the anterior portion of the superior sagittal sinus with Braun intravenous cannulas. Then, we laparatomized the rats to cannulate the portal vein, inferior caval vein, and abdominal aorta for their respective pressure recordings.

Notably, normal rats exhibited a superior sagittal sinus pressure of −24 to −27 mmHg and portal pressure of 3–5 mmHg, which was similar to that of the inferior caval vein, although with at least 1 mmHg higher values in the portal vein. By contrast, the abdominal aorta blood pressure values at the level of the bifurcation were 100–120 mm Hg [2,3,4,5,6,7,8,9].

*Brain volume and vessel presentation.* Brain volume and vessel presentation were proportional with the change in the surface area of the brain or vessel. We used the protocol previously described [4,6,7,8,9]. At 3 h after each of the sessions, the presentations of the brain and peripheral veins (superior mesenteric, inferior caval, and azygos veins) were recorded in deeply anaesthetized rats, with a camera attached to a VMS-004 Discovery Deluxe USB microscope (Veho, Dayton, OH, USA), before the procedure in control rats and just before sacrifice in rats administered lithium. The borders of the brain or veins in the photographs were marked using ImageJ computer software. Then, the surface area (in pixels) of the brain or veins was measured using a measuring function. This was performed with brain photographs before the application and at intervals after the application for both control and treated animals. In the rats with occluded mesenteric veins, the surface area of the brain or vein before application was marked as 100%, and the ratio of each subsequent brain area to the first area was calculated ($\frac{A_{2}}{A_{1}}$). Using the square-cube law shown in Equations (1) and (2), an equation for the change in brain volume proportional to the change in the surface area of the brain (6) was derived. In expressions (1)–(5), *l* is defined as any arbitrary one-dimensional length of the brain (for example, the rostro-caudal length of the brain) and is used only to define the one-dimensional proportion (l_2_/l_1_) between two observed brains and as an inter-factor (and, therefore, was not measured [6]) to derive the final expression (6). The procedure was as follows:(1)$$A_{2}=A_{1}\times {\left(\frac{l_{2}}{l_{1}}\right)}^{2}$$

(Square-cube law), $V_{2}=V_{1}\times {\left(\frac{l_{2}}{l_{1}}\right)}^{3}$; (2) (Square-cube law), $\frac{A_{2}}{A_{1}}={\left(\frac{l_{2}}{l_{1}}\right)}^{2}$; (3) (from (1), after dividing both sides by A_1_),$\frac{l_{2}}{l_{1}}=\sqrt{\frac{A_{2}}{A_{1}}}$; (4) (from (3), after taking the square root of both sides), $\frac{V_{2}}{V_{1}}={\left(\frac{l_{2}}{l_{1}}\right)}^{3}$; (5) (from (2), after dividing both sides by V_1_), $\frac{V_{2}}{V_{1}}={\left(\sqrt{\frac{A_{2}}{A_{1}}}\right)}^{3}$; and (6) (after incorporating expression (4) into Equation (5)).

*Stomach lesions.* The presentations of the gross lesions in the gastrointestinal tract were recorded in deeply anaesthetized rats, with a camera attached to a VMS-004 Discovery Deluxe USB microscope (Veho, USA). At 3 h after each of the sessions, we assessed the hemorrhagic congestive lesions in the stomach (sum of the longest diameters (in mm)).

*Microscopy. Tissue preparation.* The brain, liver, kidney, lungs, heart, stomach, intestines, and quadriceps muscle tissues were fixed in 10% neutral buffered formalin (pH 7.4) at room temperature for 24 h. Representative tissue specimens were dehydrated and embedded in paraffin, sectioned at 4 μm, and stained with hematoxylin-eosin according to the following automated Sakura Tissue-Tek DRS 2000 Slide Stainer protocol (https://www.sakura.eu/Solutions/Staining-Coverslipping/H-E-Kit accessed on 19 October 2021): rehydration in distilled water, staining with hematoxylin, washing in running tap water, differentiation with 70% alcohol, staining with eosin, dehydration, clearing, and mounting. Tissue injury was evaluated microscopically by two blinded examiners (board-certified pathologists, A.S. and E.L.) using an Olympus BX51 microscope and an Olympus 71 digital camera for saving images as uncompressed 24-bit RGB TIFF files.

*Brain histology.* Brain injury in different regions [4,6,7,8,9,55] was evaluated using a semiquantitative neuropathological scoring system as described [4,6,7,8,9,56] (Table 1), providing a common score 0–8 (grade 0 indicates no histopathologic damage).

*Lung histology.* A scoring system to grade the degree of lung injury was used in lung tissue analysis. Features were focal thickening of the alveolar membranes, congestion, pulmonary edema, intra-alveolar hemorrhage, interstitial neutrophil infiltration, and intra-alveolar neutrophil infiltration. Each feature was assigned a score from 0 to 3 based on its absence (0) or presence to a mild (1), moderate (2), or severe (3) degree, and a final histology score was determined [4,6,7,8,9,57].

*Renal, liver, and heart histology.* The criteria for renal injury were based on degeneration of Bowman’s space and glomeruli, degeneration of the proximal and distal tubules, vascular congestion, and interstitial edema. The criteria for liver injury were vacuolization of hepatocytes and pyknotic hepatocyte nuclei, activation of Kupffer cells, and enlargement of sinusoids. Each specimen was scored using a scale ranging from 0 to 3 (0: none, 1: mild, 2: moderate, and 3: severe) for each criterion, and a final histology score was determined [4,6,7,8,9,58]. Myocardial injury features used in analyzing heart lesions were based on the severity of congestion (each specimen was scored using a scale ranging from 0 to 3 (0: none, 1: mild, 2: moderate, and 3: severe), and a final histology score was determined) and the presence or absence of myocardial infarction.

*Intestinal histology.* A histologic scoring scale adapted from Chui et al. [4,6,7,8,9,59] was used for tissue scoring on a scale of 0 to 5 (normal to severe) in three categories (mucosal injury, inflammation, and hyperemia/hemorrhage) for a total score of 0 to 15, as described by Lane et al. [4,6,7,8,9,60]. The morphologic features of mucosal injury were based on different grades of epithelia lifting, villi denudation, and necrosis; grades of inflammation were graded from focal to diffuse according to lamina propria infiltration or subendothelial infiltration; and hyperemia/hemorrhage was graded from focal to diffuse according to lamina propria or subendothelial localization.

*Muscle histology.* Transverse sections of the quadriceps muscle were used for histological evaluation and examined in a blinded fashion. A special software program, ISSA Network Station Version 4.0. (VAMSTEC, Zagreb, Croatia), was used for morphometric analysis. Five high-power fields from the quadriceps muscle, which were examined as semi-serial muscle sections, were randomly selected for analysis. In the selected areas, the smallest diameters of the smallest muscle fibers were measured as previously described, and the healthy values of the quadriceps muscle (31 ± 3 mm) were considered normal [53,61,62,63].

*Oxidative stress.* At the end of the experiment, at 3 h after each session, oxidative stress in the collected tissue samples (brain, heart, lung, liver, kidney, and quadriceps muscle) was assessed by quantifying the thiobarbituric acid-reactive species (TBARS) as malondialdehyde (MDA) [34,35,36]. The tissue samples were homogenized in PBS (pH 7.4) containing 0.1 mM butylated hydroxytoluene (BHT) (TissueRuptor, Qiagen, Valencia, CA, USA) and sonicated for 30 s in an ice bath (Ultrasonic Bath, Branson, MI, USA). Trichloroacetic acid (TCA, 10%) was added to the homogenate, the mixture was centrifuged at 3000 rpm for 5 min, and the supernatant was collected. Then, 1% TBA was added, and the samples were boiled (95 °C, 60 min). The tubes were then kept on ice for 10 min. Following centrifugation (14,000 rpm, 10 min), the absorbance of the mixture was determined at the wavelength of 532 nm.

The concentration of MDA was read from a standard calibration curve plotted using 1,1,3,3-tetraethoxypropane (TEP). The extent of lipid peroxidation was expressed as MDA using a molar extinction coefficient for MDA of 1.56 × 10^5^ mol/L/cm. The protein concentration was determined using a commercial kit. The results are expressed in nmol/g of protein.

*Lithium analysis*. At 20 min or 3 h after administration of lithium sulfate (500 mg/kg ip) (lithium-time), followed by BPC 157 10 ng/kg ip, or saline 5 mL/kg ip, the rats were euthanized and blood and tissue samples taken for lithium analysis, i.e., serum, brain, muscle, heart, stomach, spleen, kidney, liver, and intestine. Blood samples were collected in vacutainer tubes without anticoagulant (BD Vacutainer Trace Element Serum, Ref 368380) (Becton-Dickinson, Franklin Lakes, NJ, USA), centrifuged at 3000 rpm for 15 min, and serum transferred into 2 mL CryoPure Tubes (Sarstedt, Nümbrecht, Germany) was kept at −20 °C until analysis. Lithium in serum and tissue samples was quantified by inductively coupled plasma mass spectrometry (ICP-MS) using Agilent 7500 cx (Agilent Technologies, Tokyo, Japan). Tissue samples were prepared for the analysis by microwave-assisted digestion in 75% (*v/v*) HNO_3_ in an UltraCLAVE IV (Milestone, Italy) microwave following the procedure detailed elsewhere (Vihnanek Lazarus 2013). After digestion, samples were adjusted to 6 g with ultrapure water (GenPure, TKA System GmbH, Niederelbert, Germany), and additionally diluted to 1:10 with 1% (*v/v*) HNO_3_ and 3 µg/L internal standards before analysis. Serum samples were diluted 1:20 with a solution containing 0.7 mM ammonia, 0.01 mM EDTA, 0.07% Triton X-100, and 3 µg/L of internal standards in ultrapure water. The blanks, matrix-matched calibration standards, and reference materials were prepared in the same manner as the samples. Single-element standard solutions (1000 ± 7 mg/L) used for calibration (Li) and as internal standards (Ge, Rh, Tb and Ir, Lu) were obtained from SCP SCIENCE (SCP Science, QC, Canada). To confirm the accuracy of the measurements, the following reference materials in serum/plasma were used: ClinChek^®^ Serum Controls (Levels I and II), ClinChek^®^ Plasma Controls (Levels I and II), (Recipe, Munich, Germany), and SeronormTM Trace Elements Serum (Levels I and II), (Sero AS, Billingstad, Norway). Overall recoveries were in the range of assigned analytical values. Previously reported concentrations of lithium in the rat plasma from 3 month old female and male Wistar rats were 6.5 ± 0.73 µg/L and 5.8 ± 0.28 µg/L, respectively [64].

### 2.4. Statistical Analysis

Statistical analysis was performed by parametric one-way analysis of variance (ANOVA), with the post-hoc Newman–Keuls test, non-parametric Kruskal–Wallis test, and subsequent Mann–Whitney U test to compare groups. Values were presented as the mean ± standard deviation (SD) and as the minimum/median/maximum. To compare the frequency difference between groups, the chi-square test or Fischer’s exact test was used. *p* < 0.05 was considered statistically significant.

## 3. Results

We revealed stable gastric pentadecapeptide BPC 157 to be an effective therapy against high-dose lithium regimens, as an occlusion-like syndrome comparable to the syndromes that follow occlusion of the major central or peripheral vessel in rats, and consequent multiorgan dysfunction syndrome. The syndrome occurred centrally and peripherally (in the form of blood pressure disturbances (intracranial (superior sagittal sinus) hypertension, portal and caval hypertension, and aortal hypotension), thrombosis, and ECG disturbances) (Figure 1 and Figure 2). Peripherally, the syndrome was characterized by constant vein congestion (i.e., superior mesenteric vein and inferior caval vein) and failure (azygos vein) (Figure 3), failed presentation of the bypassing loops (the azygos vein had almost disappeared) (Figure 3), and multi-organ lesions: heart (Figure 4), lung (Figure 5), liver (Figure 6), kidney (Figure 7), and gastrointestinal (Figure 8). Occurring centrally were brain swelling, increased brain volume, and brain lesions in all four investigated areas: cortex, hippocampus, hypothalamus, and thalamus (Figure 9, Figure 10, Figure 11, Figure 12, Figure 13, Figure 14, Figure 15, Figure 16 and Figure 17). Increased MDA levels confirmed the damage to the tissues (Figure 18). Decreased muscle fiber diameter and muscle weakness (i.e., the limited amount of time the rats could hold on to the grid) (Figure 19) may indicate increased fatigue and failed muscles. These disturbances were all counteracted by the application of the stable gastric pentadecapeptide BPC 157.

### 3.1. A Perilous Syndrome Occurred Centrally and Peripherally

#### 3.1.1. Portal and Caval Veins, Abdominal Aorta, and Superior Sagittal Sinus Pressure Recordings

Without BPC 157 therapy, the rats who were administered lithium continuously exhibited severe intracranial, portal, superior mesenteric, and caval hypertension (portal hypertension exceeding caval hypertension), and aortal hypotension. As seen after the first session, lithium administration replaced the normal (negative) pressure in the superior sagittal sinus with an increased (positive) pressure (Figure 1a–d). With BPC 157 therapy (in µg and ng regimens), the severe portal, superior mesenteric, and caval hypertension, and aortal hypotension, were rapidly attenuated or resolved. The increased (positive) pressure in the superior sagittal sinus was reverted to its normal negative pressure.

#### 3.1.2. Thrombosis

In the rats administered lithium, thrombosis consistently appeared in veins and arteries, as illustrated by thrombosis in the superior mesenteric artery and vein (Figure 1e,f). BPC 157, given after lithium, markedly counteracted this thrombosis.

#### 3.1.3. ECG Recordings

ECG recordings in the rats administered lithium, without concomitant BPC 157 therapy, regularly showed significant ST elevation, prolonged QTc intervals, and atrioventricular conduction disturbances (i.e., total AV block), in addition to marked bradycardia. By comparison, in BPC 157-treated rats, there were no repolarization changes that had been noted in the control group. Additionally, the conduction system of the heart functioned normally and the heart frequency was normal at all time checkpoints, without any atrioventricular conduction disturbances (Figure 2).

### 3.2. Perilous Syndrome Occurred Peripherally

#### 3.2.1. Vein Congestion

The proportional change in the vein area was used for the assessment of the recordings of the failure to develop the peripheral vessels (Figure 3).

This effect was consistent with the effect on the failure of peripheral blood vessels (control rats) or the recovery from the failure of peripheral blood vessels (BPC 157). The rats who received the lithium regimens, unless BPC 157 therapy was given, rapidly developed failure of the peripheral blood vessels (proportional with the change in the vein surface area).

The initial vein congestion (i.e., inferior caval and superior mesenteric veins) or vein failure (azygos vein) induced with the lithium regimens, without BPC 157 therapy, in the control rats that received saline in addition, remained constant in all of the assessed intervals.

With BPC 157 therapy, in the activated vein pathway representing circumstances in the corresponding organs (i.e., heart, lung, and liver), the presentation of the veins was reversed in a particular way. With concomitant BPC 157 therapy, the reversal of the congested vessels to non-congested vessels showing blood flow (close to healthy presentation) (superior mesenteric vein and inferior caval vein; volume control ˃ volume BPC 157) was observed. With BPC 157 therapy, the function of the vessels was restored (i.e., the azygos vein) and, thereby, volume control ˂ volume BPC 157. These may provide the particularly activated pathways between the inferior caval vein and left superior caval vein, with the presentation of the reorganized blood flow compensating for the lithium-induced effect. In addition, these findings indicated persistent action and maintained vessel integrity, even against advanced lithium toxicity, with subsequent lithium administration.

#### 3.2.2. Heart, Lung, Liver, Kidney, and Gastrointestinal Lesions

As a result of the administration of lithium sulfate, marked lesions appeared in the heart (Figure 4), lung (Figure 5), liver (Figure 6), kidney (Figure 7), and gastrointestinal tract (Figure 8) of the rats administered lithium. These lesions were markedly attenuated or even eliminated with concomitant BPC 157 therapy.

##### Heart Damage

Rats treated with lithium sulfate for 3 days that received saline medication intraperitoneally since the beginning of treatment presented with severe myocardial congestion, along with subendocardial infarcts and neutrophilic infiltration of the infarcted areas, in particular after the second and third doses of lithium (Figure 4). Contrarily, all rats who were treated with lithium and concomitantly received BPC 157 therapy exhibited no change within the myocardium (*p* ˂ 0.05, at least, vs. control, Fisher’s exact probability test, considering myocardial infarct).

##### Lung Damage

Rats treated with lithium sulfate for 3 days that received saline medication intraperitoneally since the beginning of the treatment presented with marked lung congestion and pulmonary edema, which led to progressive focal intra-alveolar hemorrhage and focal interstitial neutrophil infiltration with subsequent doses of lithium. Contrarily, all lithium-treated rats that received concomitant BPC 157 therapy exhibited only discrete lung edema and congestion with no additional changes in lung parenchyma (Figure 5).

##### Liver Damage

Rats treated with lithium sulfate for 3 days that received saline medication intraperitoneally since very beginning of the treatment presented with severe vascular congestion within the liver parenchyma, with dilatation of vascular channels in the portal tracts, central veins, and sinusoids (Figure 6). Contrarily, all lithium-treated rats that received concomitant BPC 157 therapy presented with normal liver parenchyma.

##### Kidney Damage

Rats treated with lithium sulfate for 3 days that received saline medication intraperitoneally since the beginning of treatment presented with severe vascular congestion and mild interstitial edema in the renal parenchyma and epithelial degeneration of the proximal tubules, particularly after the second and third doses of lithium (Figure 7). Contrarily, all lithium-treated rats that received concomitant BPC 157 therapy presented with normal renal parenchyma.

##### Gastrointestinal Lesions

Without BPC 157 therapy, lithium-treated rats regularly showed hemorrhagic lesions in the stomach macroscopically (Figure 8) and congestion in the gastrointestinal tract microscopically (Figure 9). Contrarily, BPC 157-treated rats presented with markedly fewer mucosal lesions in the stomach macroscopically and a normal gastrointestinal tract microscopically.

### 3.3. Perilous Syndrome Occurred Centrally

#### 3.3.1. Brain Swelling and Counteraction

The proportional change in the brain surface area was used for the assessment of the brain-swelling recording (Figure 10, Figure 11 and Figure 12).

Immediately after lithium sulphate administration, the rats rapidly developed brain swelling (brain volume proportional with the change in the brain surface area revealed an immediate increase to 120% of the healthy presentation), with a timely progression (Figure 10). 

By comparison, in the lithium-induced swollen brain, BPC 157 therapy rapidly attenuated the lithium-induced brain swelling, showing a consistent and prominent effect in returning the brain presentation to close to the normal, pre-procedure values, as a result of µg- and ng-regimens (Figure 10). Similarly, an apparent counteraction of the brain swelling was seen with BPC 157 regimens in the lithium course, after each of the three lithium administrations (Figure 11 and Figure 12).

#### 3.3.2. Brain Damage

After each of the three lithium administrations, there was marked intracerebral hemorrhage in the fronto-parietal area. The control lithium rats had a larger hemorrhage area affecting the deeper brain, and hemorrhaging was found not only in the grey matter, but also in the white matter. The intracerebral hemorrhage affecting the same area in BPC 1570-treated rats was smaller in diameter, and affected only the grey matter (Figure 13). Despite acute, hemorrhagic features within the brain tissue, the BPC 157 therapy after lithium regimens resulted in a considerable reduction in lesions. Only a mild edema and congestion was observed. No or few karyopyknotic neuronal cells in the analyzed neuroanatomic structures were found, primarily in the hippocampus, after the second and third lithium administrations (Figure 14, Figure 15, Figure 16 and Figure 17). By comparison, the control lithium-treated rats presented severe neuropathological changes in the central nervous system, and a generalized increase in edema and congestion. An increased number of karyopyknotic cells were found in all four regions: cerebral and cerebellar cortex, hypothalamus/thalamus, and hippocampus. In particular, there was evidence of karyopyknosis and degeneration of Purkinje cells of the cerebellar cortex, and a progression from mild to marked karyopyknosis of cortical neurons and pyramidal cells of the hippocampus (Figure 14, Figure 15, Figure 16 and Figure 17).

### 3.4. Oxidative Stress

Without medication, rats regularly showed increasedmalondialdehyde (MDA) values with the ligated superior mesenteric vein (Figure 18). This was completely counteracted in the rats that received BPC 157 medication.

### 3.5. Muscular Weakness

Lithium rats presented a decrease in the diameter of the muscle fibers and muscle weakness. They held on to the grid for a very limited amount of time, as a reflection of fatigue and muscle weakness, after the first, second, and third lithium administrations. These disturbances were counteracted in lithium-treated rats that received BPC 157 regimens (Figure 19).

### 3.6. Lithium Serum Analysis

At the early time (i.e., 20 min lithium-time), the presentation of early lesions in the lithium-overloaded rats presented a strict difference, namely, the regular injury course in the control-lithium rats, and a markedly attenuated course in the lithium-BPC 157-treated rats due to the beneficial effect of the BPC 157 therapy (see above, i.e., Figure 10 (brain swelling), Figure 19 (muscle weakness)). However, BPC 157-lithium-treated rats and saline-lithium-treated rats shared the same high serum lithium concentration. Similarly, concentrations of lithium in samples of rat organs showed relatively similar values in the brain, heart, liver, spleen, muscle, stomach, and intestine. The concentrations of lithium in the samples of kidney were higher in the BPC 157-lithium-treated rats (Figure 20). A similar situation was seen at the later time (i.e., 3 h lithium-time). There was also an advanced lithium course (i.e., lower concentration in the serum, higher in the brain); the severity of the lesions obviously differed in the lithium-overloaded rats (saline-rats vs. BPC 157-rats), but all of the rats had similar lithium concentrations in the serum. Similar lithium concentrations were also noted in the organs (with the exception of the stomach, which showed a lower concentration in the lithium-BPC 157-treated rats) (Figure 20).

In summary, the evidence showed that, in the lithium-treated rats with three subsequent lithium administrations, BPC 157 therapy rapidly attenuated the severe portal and caval hypertension and aortal hypotension. This may be due to the rapid recruitment of collateral vessels (i.e., the activated azygos vein), which also rapidly eliminates the increased pressure in the superior sagittal sinus. Thus, there is a rapid and adequate resolution of the anatomical imbalance in venous drainage, which acts both peripherally and centrally. The attenuation of brain, heart, lung, liver, kidney, and gastrointestinal lesions, thrombosis, and muscle weakness may be an automated result.

## 4. Discussion

In this novel lithium demonstration, lithium occlusive-like syndrome appeared to be a consequence of high-dose lithium-induced endothelial impairment [1]. We showed that the lithium toxicity in rats presents as an occlusive-like syndrome, and the injurious regimen with high-dose lithium (once daily for 3 consecutive days) corresponds to the vessel occlusion syndromes [2,3,4,5,6,7,8] and intragastric absolute alcohol-induced occlusion-like syndrome [9], thereby sharing the symptoms of intracranial, portal, and caval hypertension, aortal hypotension, and multi-organ dysfunction syndrome. Consequently, the beneficial effects of concomitant BPC 157 therapy may prevent all adverse effects of the lithium occlusion-like syndrome, vessel occlusion syndromes [2,3,4,5,6,7,8], and intragastric absolute alcohol-induced occlusion-like syndrome [9].

This combination may explain the noted rapid full presentation of the general toxicity of lithium in rats, the entire peripheral and central syndrome, and progression with the next dose. The rapidity of the combination thereby occurs consequent to the injurious event induced by the high lithium dose, and blood vessel function failure, such as that induced by direct vessel occlusion [2,3,4,5,6,7,8]. It is likely that, in the case of lithium, this indicates irremovable (endothelial) lesions [1], in which the full peripheral and central syndromes appear in addition to each other, as also noted with intragastric absolute alcohol-induced occlusion-like syndrome [9]. Due to the prime peripheral or central lesion, these lesions should appear as an essential cause–consequence cycle. This venous and intracranial hypertension, or increased intra-abdominal (and intrathoracic) pressure, is rapidly transmitted through the venous system and increases intracranial pressure [65,66]. Conceptually, this may be due to brain swelling and intracranial (superior sagittal sinus) hypertension; brain lesions; portal and caval hypertension; aortal hypotension; widespread thrombosis in veins and arteries; congestion of the superior mesenteric and inferior caval veins; failure of the azygos vein; ECG disturbances; and heart, lung, liver, kidney, and gastrointestinal lesions. Thus, in the case of lithium, the pleiotropic beneficial effects of BPC 157 therapy, which is known to rapidly attenuate or eliminate the consequences of the irremovable occlusion of various vessels, both peripherally and centrally [2,3,4,5,6,7,8], may have considerable conceptual importance. In addition, the similar occlusive-like syndrome induced by absolute alcohol intragastric instillation, and the similar therapy effect of BPC 157 administration, reveal a shared endothelial impairment and vascular failure that has been previously addressed [9]. Together, these are all included in the general syndrome caused by the high-dose lithium regimen, and should be entirely counteracted as proof of the concept of the essential interaction with innate lithium activity and tissue damage. This point was fully exemplified, from the outset, by the beneficial effects of the application of BPC 157.

Notably, in the early treatment time (i.e., 20 min lithium-time), lithium-overloaded BPC 157 rats with an apparently attenuated lithium-course shared the same high serum lithium concentration as the lithium-overloaded control injury course (for illustration, see Figure 10 (brain swelling and BPC 157 effect), and Figure 19 (muscle weakness and BPC 157 effect). This is likely indicative of either further deterioration (lithium-overloaded control rats) or an attenuated lithium course (lithium-overloaded BPC 157 rats). A similar situation appeared with the advanced lithium course (at 3 h lithium-time; lesser lithium concentration in the serum, higher concentration in the brain [17]). In addition, these BPC 157 effects may reveal an intriguing point in relation to normal functioning with high serum lithium, rather than the disturbed function that may be expected [17]. This beneficial effect of BPC 157 can essentially replicate a strong innate counteraction, with a potential life-saving effect. Such an effect has also been noted with constant severe and eventually lethal hyperkalemia (in which the lethal effect, severe arrhythmias, and muscle disability were counteracted while serum potassium concentration remained high) after an intraperitoneal potassium chloride over-dose challenge [52]. Note, considering the lithium-endothelium effect [1], potassium has also been shown to cause endothelial damage [9]. It is possible that the high serum (lithium and potassium) values are suggestive of an immediate (and continuous) vascular failure, and a consequent tissue damaging effect, which may be counteracted, immediately and continuously, by BPC 157 therapy. However, it is evident that such a lithium over-load (the same high serum levels and low brain levels at an early interval, and low serum levels and high brain levels at a late interval [17]), excludes the possibility that the relatively good results in the treatment group were due to a lower drug concentration in the blood. Finally, in patients, aversion to lithium can be elicited even when serum lithium levels are within the “therapeutic” range [17,67].

Thus, in the lithium-treated rats, the brain swelling and increase in intracranial (superior sagittal sinus) hypertension may be the cause or the consequence of the severe brain lesions in all areas. An increased number of karyopyknotic cells were found in all four brain regions—cerebral and cerebellar cortices, hypothalamus/thalamus, and hippocampus. Moreover, increased edema and congestion in areas of the cerebral cortex were revealed, particularly karyopyknosis and degeneration of Purkinje cells in the cerebellar cortex and pyramidal cells of the hippocampus. Thus, lithium-treated rats exhibited a harmful inability to adequately drain venous blood for a given cerebral blood inflow without raising venous pressures, thereby leading to sudden venous and intracranial hypertension [68,69]. However, this may be initiated either centrally or peripherally, providing the similar noxious course observed in the rats with occlusion of the superior sagittal sinus [8], and in the rats with occlusion of the superior mesenteric artery and/or vein [4,6,7]. Furthermore, as in the case of rats with major vessel occlusion [2,3,4,5,6,7,8], in addition to absolute alcohol intragastric instillation [9], the lithium-treated rats without concomitant BPC 157 therapy experienced severe brain injuries and exhibited considerable heart and lung disturbances (i.e., prolonged QTc-interval, subendocardial infarct, heart dysfunction, and congestion and hemorrhage in the lung parenchyma (resembling the exudative features of ARDS)). Then, as noted in the peripheral and central vessel occlusion syndromes [2,3,4,5,6,7,8] and absolute alcohol intragastric instillation occlusion-like peripheral and central syndrome [9], the lithium-treated rats consequently exhibited both liver failure and kidney failure, progression of congestion, and extensive gastric hemorrhagic lesions, in addition to prominent portal and caval hypertension, and congestion of the inferior caval and superior mesenteric veins. Escalating thrombosis is a shared common point [2,3,4,5,6,7,8,9], reflecting general stasis (i.e., large volumes trapped in the damaged stomach, CNS, and portal and caval vein tributaries, which may also perpetuate the brain and heart ischemia) and failed activation of the collateral bypassing pathways. This rapidly appeared within minutes, while the major veins (inferior caval and superior mesenteric veins) appeared to be congested and the azygos vein failed. Furthermore, lithium may induce direct damage to the liver, kidneys [70,71,72,73,74], heart [75,76], lungs [77,78], and brain [79,80].

Similarly, as in the case of rats with an occluded peripheral or central vessel [2,3,4,5,6,7,8], in addition to absolute alcohol intragastric instillation occlusion-like peripheral and central syndrome [9], BPC 157 therapy is also effective in lithium-treated rats. Thus, the similar therapeutic effect illustrates the beneficial effects of administration of BPC 157 in lithium-treated rats. This treatment rapidly attenuated brain swelling and intracranial (superior sagittal sinus) hypertension. Note, in the rats with occlusion of the superior sagittal sinus, BPC 157 was shown to maintain normal (negative) pressure values, even when confronted with additional challenges (volume application) that originated within the cranium or in the periphery, given a cranial or peripheral intravenous challenge [8]. These findings correlate with the counteracted brain swelling in lithium-treated rats due to the rapid effect of BPC 157 therapy. Consequently, lithium-treated rats receiving concomitant treatment with BPC 157 exhibited only a few karyopyknotic neuronal cells (primarily in the hippocampus). Evidently, the counteracted elevated superior sagittal sinus pressure and brain lesions, as a result of the application of BPC 157, emphasizes its ability to adequately drain venous blood for a given cerebral blood inflow. As such, this effect may be the cause or the consequence of a number of characteristics, including no changes within the myocardium or renal parenchyma, counteracted ECG disturbances, only mild lung and liver congestion, eliminated portal and caval hypertension, markedly attenuated aortal hypotension, abrogated hemorrhagic stomach lesions, and almost total elimination of venous and arterial thrombosis, thereby counteracting stasis, as ascertained by more adequate blood flow. In this regard, the activated azygos vein pathway, more directly combining left superior vein and inferior caval vein tributaries (while in the control rats, the azygos vein completely failed), may be analogous to the findings in vessel occlusion studies in rats with Budd–Chiari syndrome [5] or central venous occlusion [8].

For the noted BPC 157-lithium activity, the specific “bypassing key”, which is a common point in vascular occlusion studies [2,3,4,5,6,7,8], absolute alcohol intragastric instillation occlusion-like peripheral and central syndrome [9], and in the lithium-treated rats, may represent the noted special interaction with the NO system and NO agents in the various models and species [36]. BPC 157 induced the release of NO on its own [81,82], counteracted the induced hypertension and pro-thrombotic effects (L-NAME) [79,81], and induced hypotension and anti-thrombotic (L-arginine) effects [83,84]). The specific effects on blood pressure and maintenance of thrombocyte function [38,81,83,84] is in accordance with the vasomotor tone carried out through the BPC 157-specific activation of the Src-caveolin-1-endothelial nitric oxide synthase (eNOS) pathway [29]. It is possible that BPC 157 maintains the function of the prostaglandin system [37], revealing that BPC 157 counteracted the adverse effects of NSAIDs, COX-1 blockers, and COX-2 blockers [85,86,87,88,89,90]. This may also be an adjuvant arthritis therapy in rats to both prevent the development of lesions and cure already established lesions [91]. Indomethacin cytoprotection studies [27] and mitigated leaky gut syndrome revealed BPC 157 to be a stabilizer of the cellular junction, via increasing the expression of the tight junction protein ZO-1, and transepithelial resistance [27]. Findings showed inhibition of the mRNA of inflammatory mediators (iNOS, IL-6, IFNγ, and TNF-α) and increased expression of HSP 70 and 90, in addition to antioxidant proteins (HO-1, NQO-1, glutathione reductase, glutathione peroxidase 2, and GST-pi) [27]. Thus, this particular background strongly counters the particular damaging effects of lithium on vessel function, which seem to be related to the NO system [1]. BPC 157 exhibited a specific effect on the *Egr*, *Nos*, *Srf*, *Vegfr*, *Akt1*, *Plc**ɣ*, and *Kras* pathways in the vessels, providing an alternative operating pathway (i.e., the left ovarian vein as the key for the infrarenal occlusion-induced inferior caval vein syndrome in rats) [2]. Given the reperfusion in rats with stroke [25], BPC 157 therapy counteracted both the early and delayed neural hippocampal damage. In hippocampal tissues, mRNA expression studies at 1 h and 24 h showed strongly elevated (*Egr1, Akt1, Kras, Src, Foxo, Srf, Vegfr2, Nos3,* and *Nos1*) and decreased (*Nos2* and *Nfkb*) gene expression (*Mapk1* was not activated), which may be the mechanisms behind the actions of BPC 157 [25]. In the same way, along with encephalopathies [85,86,87,88,89,90], it is likely that BPC 157 counteracts multiple pathologies in the gastrointestinal tract and liver [85,86,87,88,89,90]. Moreover, in addition to the arrhythmias in rats with permanent occlusion of the major vessels [2,3,4,5,6,7,8], and absolute alcohol intragastric instillation occlusion-like peripheral and central syndrome [9], BPC 157 counteracts the various other arrhythmias [48,52,92,93,94]. In particular, BPC 157 therapy normalizes the QTc duration in rats treated with neuroleptics [48], and both prevents chronic heart failure and promotes recovery from chronic heart failure [93]. Similarly, in addition to permanent occlusion of the major vessel-induced lung lesions [2,3,4,5,6,7,8], and absolute alcohol intragastric instillation occlusion-like peripheral and central syndrome [9], BPC 157 counteracted the lung pathology that may be induced with pulmonary hypertension syndrome in chickens [95], or application of monocrotaline in rats [96] and intratracheal HCl instillation-induced lung lesions in rats [97]). Regarding the mentioned BPC 157-potassium counteracting relation [16,52,53], BPC 157 opposed potassium disturbances both in vivo (a life-saving effect in hyperkalemic rats [52]) and in vitro [52], and reduced depolarization by up to 70% compared to controlled hyperkalemic depolarization [52]. Thereby, BPC 157’s effect may be to counteract the otherwise inevitable chain of hyperkalemia events, i.e., the open sodium channels are inactivated and become refractory, increasing the threshold needed to generate an action potential [52].

In all of the tissues tested, the actions of BPC 157 against the damaging effects on blood vessel function can be also combined with an anti-oxidant effect in the lithium-treated rats administered with BPC 157 who had reduced MDA values, as a confirmative result of both preserved and rescued tissue integrity and vein integrity [2,3,4,5,6,7]. As previously described, this free radical scavenger effect occurs in both ischemic and reperfusion conditions in the various tissues (i.e., brain, colon, duodenum, cecum, liver, and veins) and plasma [2,3,4,5,6,7,21,22,23,24,39,40,41].

Furthermore, lithium also acts as a neuromuscular blockade agent [98,99,100], thereby inducing muscle weakness (pathology) and prostration in the lithium-treated rats (who were unable to hold their positions in the up-right metal grid). Similarly, considerable counteraction was provided by the administration of BPC 157, and BPC 157 is also known to counteract succinylcholine-induced muscular paralysis in rats [53]. Together, these may be taken as an additional proof of the concept. It should be noted that BPC 157 accelerated muscle healing after various injuries [61,62,63] and counteracted muscle weakness in rats with stroke [25], hyperkalemia [52], and hypermagnesemia [16], in addition to exposure to the multiple sclerosis-like neurotoxin cuprizone [101]. Moreover, as a potential agent to treat patients suffering from cancer cachexia, BPC 157 counteracted tumor-induced muscle cachexia via counteraction of several pro-inflammatory and pro-cachectic molecular pathways and mediators [28].

For the investigation of lithium therapy and its mechanisms from a number of perspectives [17], including as an occlusion-like vessel model and corresponding harmful syndrome, rapid presentation and further escalation may be conclusive evidence best defined by the observed effects themselves. This appears to be the novel combined point for further evaluation (i.e., the entire noxious syndrome, in particular, vascular failure and activation of the azygos vein collateral pathway as a form of rescue). This indicative point, although not defined more precisely, should be taken into account in the study of the direct targets, such as direct inhibition of a select group of enzymes (i.e., inositol monophosphatase (IMPase), inositol polyphosphate 1-phosphatase (IPPase), phosphoglucomutase (PGMase), biphosphate nucleotidase (BPNase), fructose 1,6-biphosphatase (FBPase), and glycogen synthase kinase-3 (GSK-3)) [17]. The same emphasis can be placed on the effects around the synapse, modulation of the neurotransmitter receptor-mediated processes, and modified production and turnover of certain neurotransmitters, particularly dopamine, glutamate, and GABA [102,103]. Specifically, these are all intended to provide insight into the ability of lithium to modulate intracellular signaling cascades, the downstream consequences of which are its effects on physiology and behavior [17]. Note, the lithium regimen used (i.e., 500 mg/kg, ip, once daily for 3 consecutive days) is higher than the usual “high” lithium dose (300 mg/kg ip) [104,105,106], but markedly below the usual LD50 for lithium application in rats [107]. This may be relevant, particularly considering the conversion of animal doses to human-equivalent doses based on body surface area [108]. By dividing the rat dose by 6.2 or multiplying by 0.16 [108], our original higher doses in rats are closer to those used in patients. Similarly, we should also consider the previously mentioned beneficial effects of BPC 157 application in particular animal behavioral models. For example, it has been shown to not only counteract the acute and chronic toxicities of amphetamine [42,43], but also neuroleptic-induced catalepsy and other neuroleptic adverse effects [44,45,46,47,48], such as L-NAME- and NOS blocker-induced catalepsy [44] and dopamine receptor supersensitivity [42]. An antidepressant effect was highlighted [49] that may counteract serotonin syndrome [50] and induce serotonin release in particular brain areas [50]. This treatment also counteracted schizophrenia-like positive symptom models through a NO system-related effect [44].

Finally, it should be highlighted that the results of animal studies per se should be interpreted cautiously, particularly those from lithium studies [17] (note, the original Cade study used a 0.5% lithium solution in guinea pigs) [109]. Similarly, there is a relative paucity of clinical data on BPC 157 [10,11,12,13,14]. Nonetheless, BPC 157 has been proven to be efficacious in the treatment of ulcerative colitis, both in clinical settings [110,111] and in experimental rats with ulcerative colitis in ischemic/reperfusion studies [21] and other ulcerative colitis models, including various species [85,86,87,88,89,90,101,112,113] and complications (for a review see [114]). A particular advantage for revealing and applying this concept in practice is its very safe profile (a lethal dose (LD1) could be not achieved) [115], which is a point recently confirmed in a large study by Xu and collaborators [116]. Finally, there is a consistently used effective range of BPC 157 (µg–ng) application and used regimens [2,3,4,5,6,7,8,9], which may support each other’s effects. Interestingly, in rats with a central venous occlusion [8], the same beneficial effect was seen in the application in the swollen brain, intraperitoneally or intragastrically. Together, these findings (for a review see [115]) may be suggestive of a physiological role (e.g., in situ hybridization and immunostaining of BPC 157 in human gastrointestinal mucosa, lung bronchial epithelium, epidermal layer of the skin, and kidney glomeruli) [115]. In this context, the role of the animal model is indispensable, and the practical indicative evidence is even more important. Furthermore, the issue of BPC 157 and lithium elaborated in this study requires additional research. This should consider the opposing beneficial effects of lithium known in the endothelium and each of the mentioned organs [117,118,119,120,121,122,123], and the evidence that lithium has been found to prevent and/or reverse DNA damage, free-radical formation, and lipid peroxidation in diverse models [124], in addition to inducing oxidative damage and inflammation [125]. However, we have revealed the toxicity of the lithium complex as an occlusive-like model that shares a pathology with syndromes after major vessel occlusion [2,3,4,5,6,7,8] or intragastric absolute alcohol administration [9]; intracranial, portal and caval hypertension; aortal hypotension; multi-organ dysfunction syndrome; and oxidative stress. We also showed the function of BPC 157 as a therapy to counteract high levels of lithium intoxication in rats.

## Figures and Tables

**Figure 1 biomedicines-09-01506-f001:**
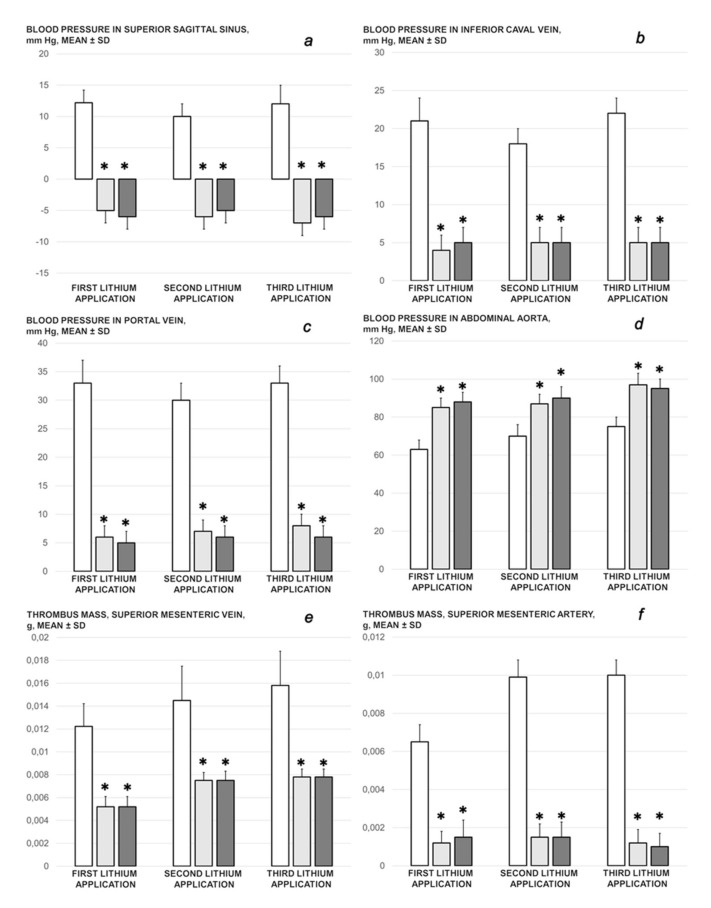
Blood pressure in the superior sagittal sinus (**a**), inferior caval vein (**b**), portal vein (**c**), and abdominal aorta (**d**), and thrombus presentation in the superior mesenteric vein (**e**) and superior mesenteric artery (**f**) in the lithium-treated rats after the first, second, and third lithium sulphate administration (500 mg/kg/day intraperitoneally for 3 consecutive days). BPC 157 10 µg/kg (light gray bars), 10 ng/kg (dark gray bars); saline 5 mL/kg (white bars) given intraperitoneally after each of the lithium administrations. Six rats/group/interval. Means ± SD, * *p* ˂ 0.05, at least, vs. control.

**Figure 2 biomedicines-09-01506-f002:**
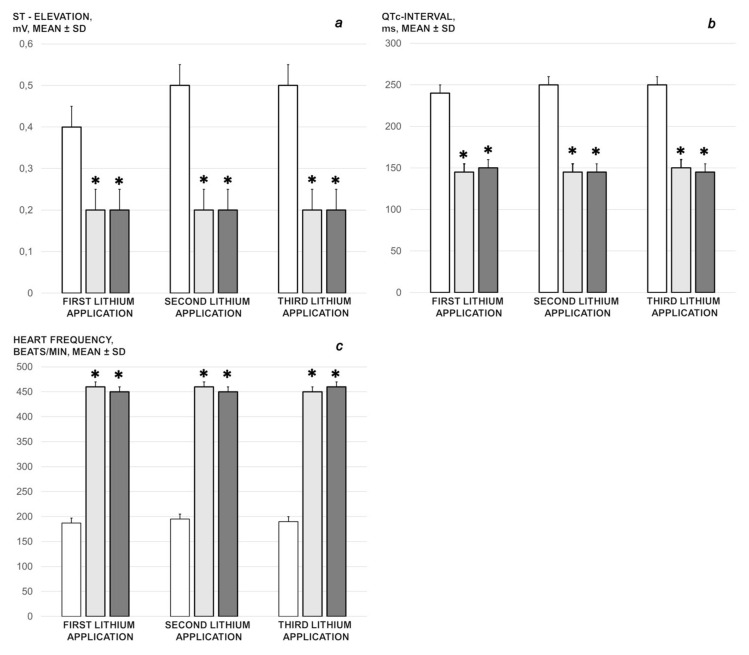
ECG changes: ST-elevation, mV, (**a**); QTc-interval prolongation, ms (**b**); heart frequency (**c**), beats/min; after first, second, and third lithium sulphate administration (500 mg/kg/day intraperitoneally for 3 consecutive days). BPC 157 10 µg/kg (light gray bars), 10 ng/kg (dark gray bars); saline 5 mL/kg (white bars) given intraperitoneally after each of the lithium administrations. Six rats/group/interval. Means ± SD, * *p* ˂ 0.05, at least, vs. control.

**Figure 3 biomedicines-09-01506-f003:**
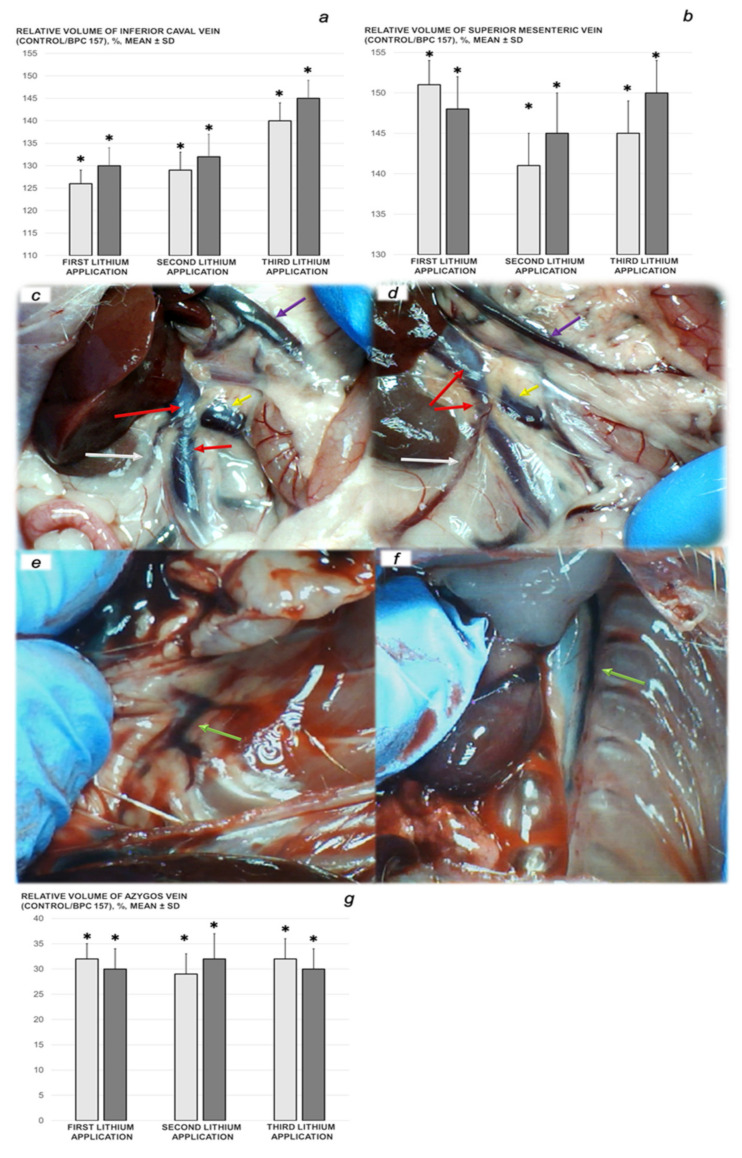
Blood vessels presentation (**a**–**g**). Relative volume of the veins (inferior caval vein (**a**), superior mesenteric vein (**b**), azygos vein (**g**), volume control/volume BPC 157, %, in the rats that received the first, second, and third lithium administration (500 mg/kg/day intraperitoneally for 3 consecutive days). BPC 157 10 µg/kg (light gray bars), 10 ng/kg (dark gray bars); saline 5 mL/kg (not shown) given intraperitoneally after each of the three lithium sulphate administrations. Six rats/group/interval. Means ± SD, * *p* ˂ 0.05, at least, vs. control (**c**–**f**). Collateral pathways (azygos vein (green arrow)) and veins congestion gross presentation in lithium-treated rats that received BPC 157 therapy (right) or saline medication (left). Inferior caval vein (red arrow) and its tributary (left) renal vein (yellow arrow), and right testicular vein (white arrow), and superior mesenteric vein (violet arrow) presentation, congested in controls (**c**) and non-congested and functioning in BPC 157-treated rats (**d**). Azygos vein presentation (**e**,**f**), almost disappeared and congested in controls (**e**), and functioning in BPC 157-treated rats (**f**).

**Figure 4 biomedicines-09-01506-f004:**
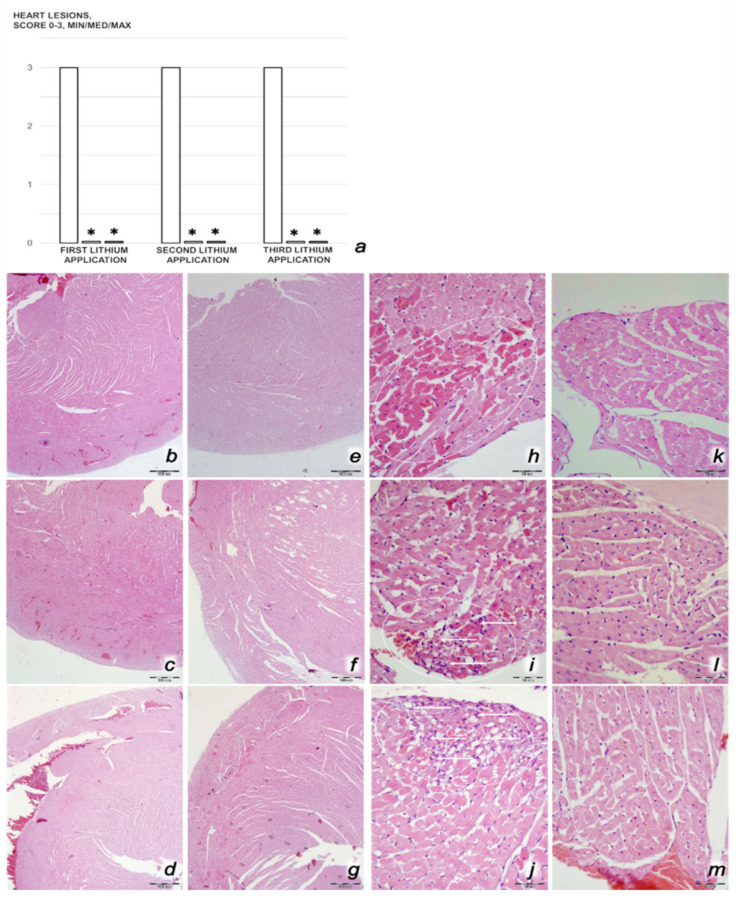
Heart pathology. (**a**) Microscopy scoring of the congestion lesions of the heart (scored 0–3) Min/Med/Max, (**a**) in the rats treated with lithium sulphate (500 mg/kg/day intraperitoneally for 3 days) that received medication (BPC 157 10 µg/kg (light gray bars), 10 ng/kg (dark gray bars); saline 5 mL/kg (white bars)) given intraperitoneally. Six rats/group/interval. * *p* ˂ 0.05, at least, vs. control. Heart histology in control lithium-treated rats (**b**–**d**,**h**–**j**) and BPC 157 lithium-treated rats (**e**–**g**,**k**–**m**); first lithium application (**b**,**e**,**h**,**k**); second lithium application (**c**,**f**,**i**,**l**); third lithium application (**d**,**g**,**j**,**m**)). A marked congestion was found in control lithium-treated rats after each of the lithium administrations. Contrarily, lithium-treated rats that received BPC 157 regimens presented normal heart presentation of minimal heart congestion. Control lithium-treated rats presented subendocardial marked ischemic myocytes (circles) after each of the lithium administrations, with degeneration and infiltration of neutrophils, particularly after the second and third lithium administrations (arrows). Contrarily, lithium-treated rats that received BPC 157 regimens presented normal myocardium (HE; magnification ×100, scale bar 100 μm (**b**–**g**); magnification ×400, scale bar 10 μm (**h**–**m**)).

**Figure 5 biomedicines-09-01506-f005:**
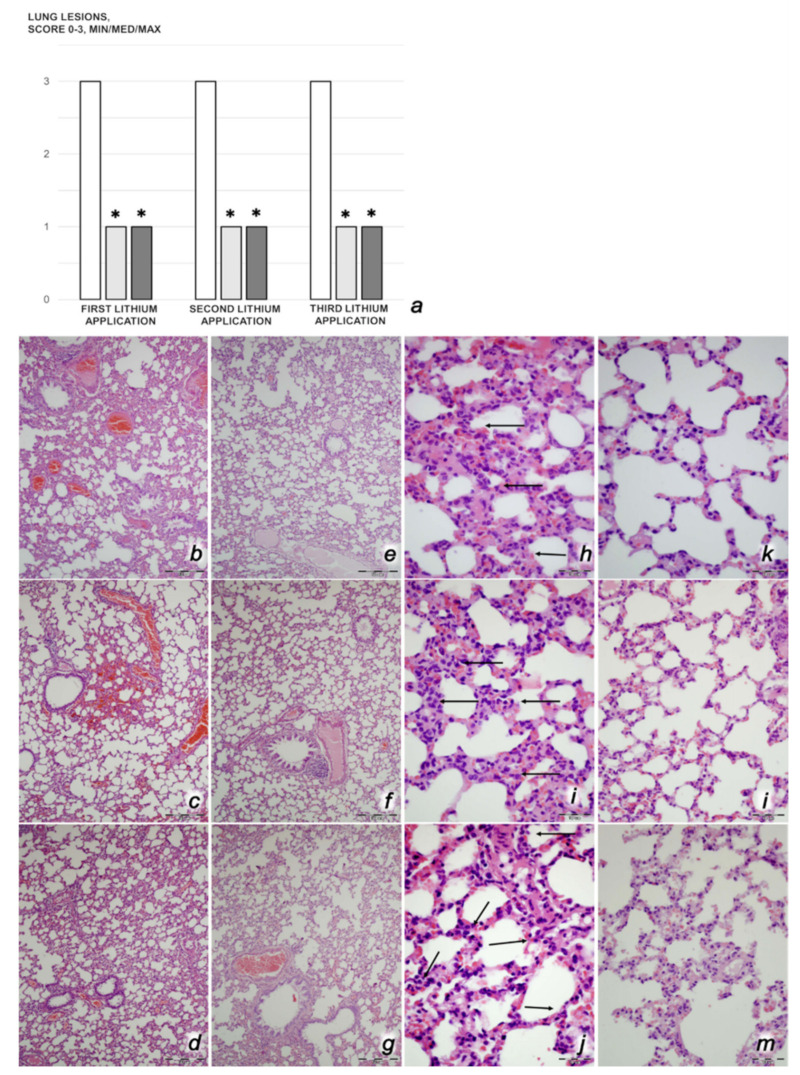
Lung pathology. (**a**) Microscopy scoring of the lesions of the lung (scored 0–3) Min/Med/Max, in the rats treated with lithium sulphate (500 mg/kg/day intraperitoneally for 3 days) that received medication (BPC 157 10 µg/kg (light gray bars), 10 ng/kg (dark gray bars); saline 5 mL/kg (white bars)) given intraperitoneally. Six rats/group/interval. * *p* ˂ 0.05, at least, vs. control. Lung histology in control lithium-treated rats (**b**–**d**,**h**–**j**) and BPC 157 lithium-treated rats (**e**–**g**,**k**–**m**); first lithium application (**b**,**e**,**h**,**k**); second lithium application (**c**,**f**,**i**,**l**); third lithium application (**d**,**g**,**j**,**m**). A marked congestion was found in controls after each of the lithium administrations, with increased appearance of intra-alveolar hemorrhage found after the second and third lithium administrations. Contrarily, lithium-treated rats that received BPC 157 regimens presented mild congestion of the lung. Interstitial neutrophil infiltration appeared in controls after each of the lithium administrations (arrows), particularly after the second and third lithium administrations. Contrarily, lithium-treated rats that received BPC 157 regimens presented thin and no inflammation within alveolar septa (HE; magnification ×100, scale bar 50 μm (**b**–**g**); magnification × 400, scale bar 10 μm (**h**–**m**)).

**Figure 6 biomedicines-09-01506-f006:**
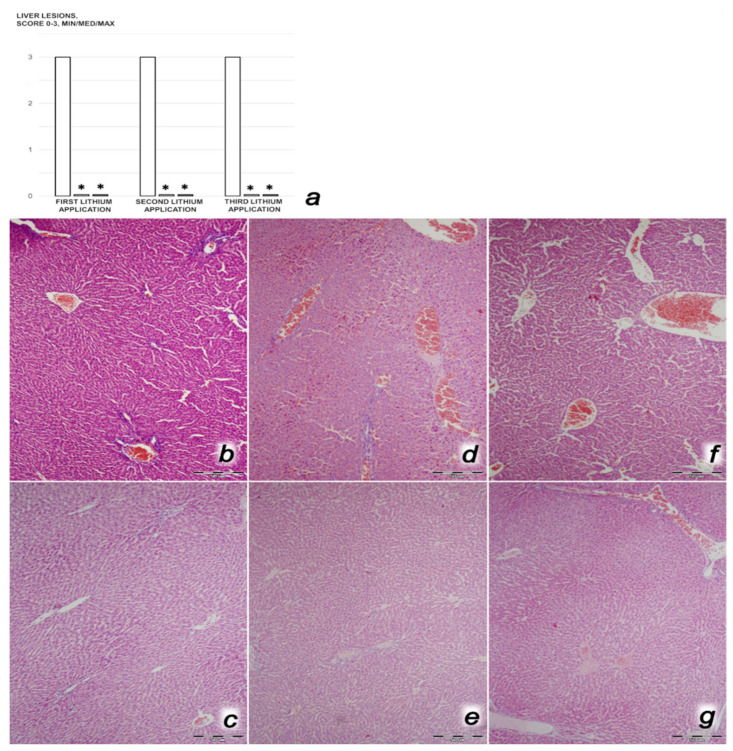
Liver pathology. (**a**) Microscopy scoring of the lesions of the liver (scored 0–3) Min/Med/Max, (**a**) in the rats treated with lithium sulphate (500 mg/kg/day intraperitoneally for 3 days) that received medication (BPC 157 10 µg/kg (light gray bars), 10 ng/kg (dark gray bars); saline 5 mL/kg (white bars)) given intraperitoneally. Six rats/group/interval. * *p* ˂ 0.05, at least, vs. control. (**b**–**g**) Illustrative histology presentation of the affected liver (**b**,**d**,**f**, control), and liver lesions counteraction (**c**,**e**,**g**, BPC 157); first lithium application (**b**, control; **c**, BPC 157), second lithium application (**d**, control; **e**, BPC 157), third lithium application (**f**, control; **g**, BPC 157). A congestion and dilatation of central veins, sinusoids and portal tracts vessels was found in controls that received lithium, unlike lithium-treated rats that received BPC 157 regimens, which presented normal parenchyma (HE; magnification × 100, scale bar 50 μm).

**Figure 7 biomedicines-09-01506-f007:**
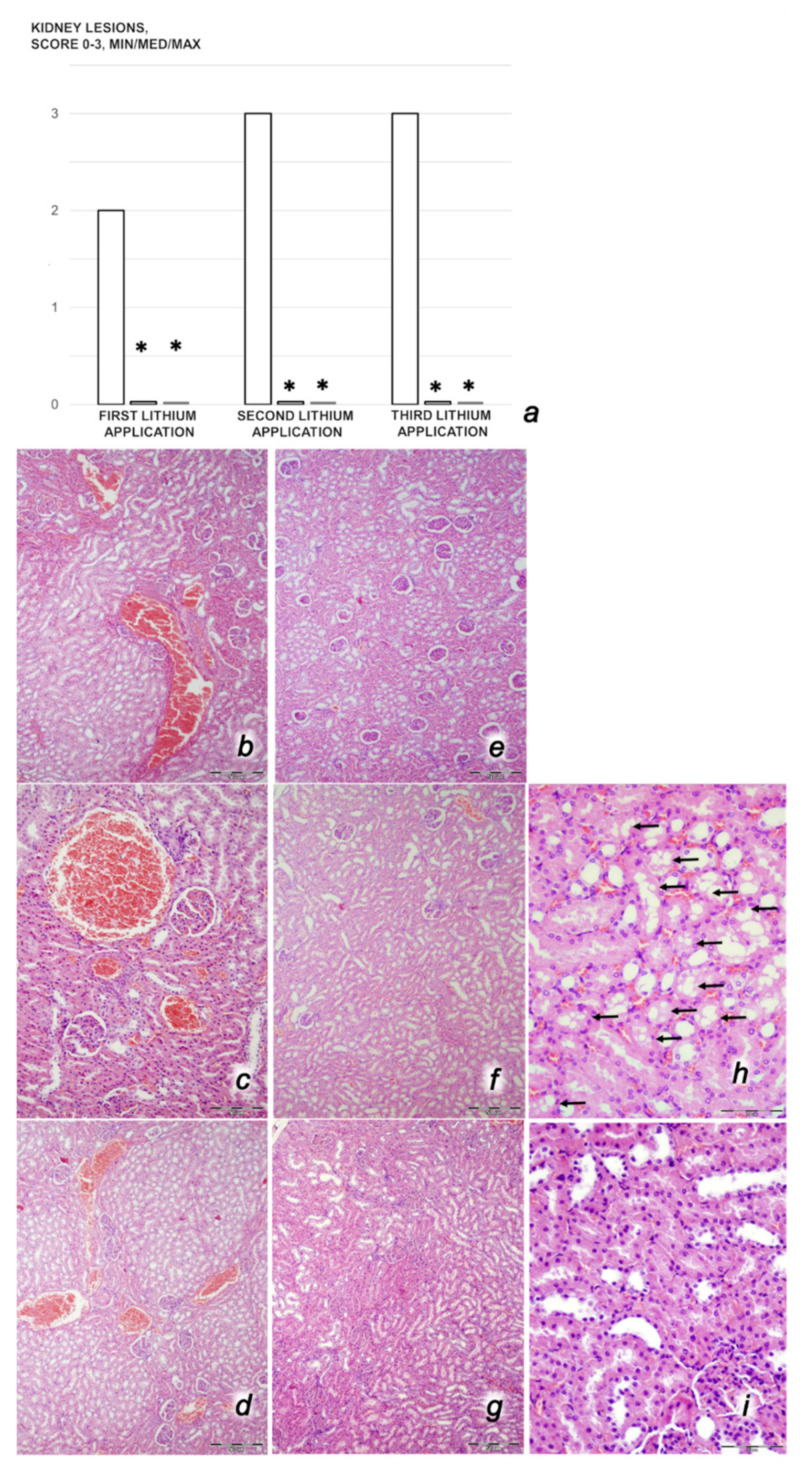
Kidney pathology. Microscopy scoring of the lesions of the kidney (scored 0–3) Min/Med/Max, (**a**) in the rats treated with lithium sulphate (500 mg/kg/day intraperitoneally for 3 days) that received medication (BPC 157 10 µg/kg (light gray bars), 10 ng/kg (dark gray bars); saline 5 mL/kg (white bars)) given intraperitoneally. Six rats/group/interval. * *p* ˂ 0.05, at least, vs. control. Kidney histology in control lithium-treated rats (**b**–**d**,**h**) and BPC 157- and lithium-treated rats (**e**–**g**,**i**); first lithium application (**b**,**e**); second lithium application (**c**,**f**); third lithium application (**d**,**g**,**h**,**i**). A marked congestion was found in controls after each of the lithium administrations. Contrarily, lithium-treated rats that received BPC 157 regimens presented normal tissue or minimal congestion. Illustrative presentation of the marked congestion and vacuolization of the renal tubular epithelia with degenerative changes in control lithium-treated rats after the third lithium administration (**h**) (arrows) and normal appearance of the renal tubules in lithium-treated rats that received BPC 157 regimens (**i**) (HE; magnification × 100, scale bar 50 μm (**b**–**g**); magnification × 400, scale bar 20 μm (**h**,**i**)).

**Figure 8 biomedicines-09-01506-f008:**
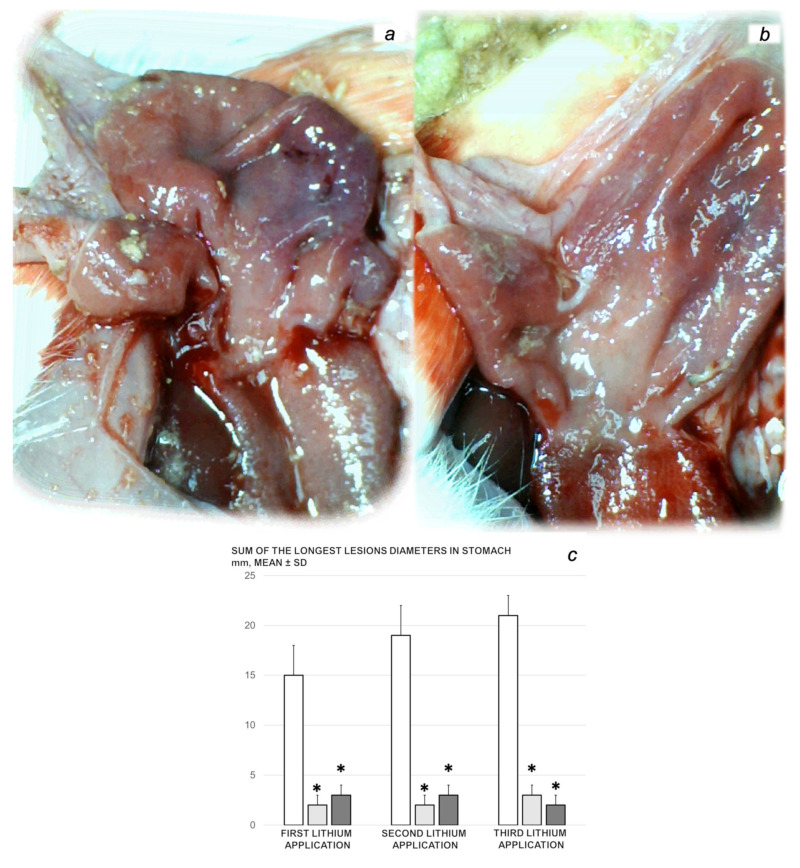
Illustrative gross gastrointestinal lesions in the stomach of the lithium-treated rats after the first lithium administration (**a**) and presentation close to normal in lithium-treated rats that received BPC 157 after lithium administration (**b**). (**c**) Mucosal lesions in the stomach (sum of the longest lesions diameters, mm, means ± SD) after the first, second, and third lithium administrations (500 mg/kg/day, intraperitoneally, for 3 consecutive days). BPC 157 10 µg/kg (light gray bars), 10 ng/kg (dark gray bars); saline 5 mL/kg (white bars) given intraperitoneally after each of the lithium administrations. Six rats/group/interval. * *p* ˂ 0.05, at least, vs. control.

**Figure 9 biomedicines-09-01506-f009:**
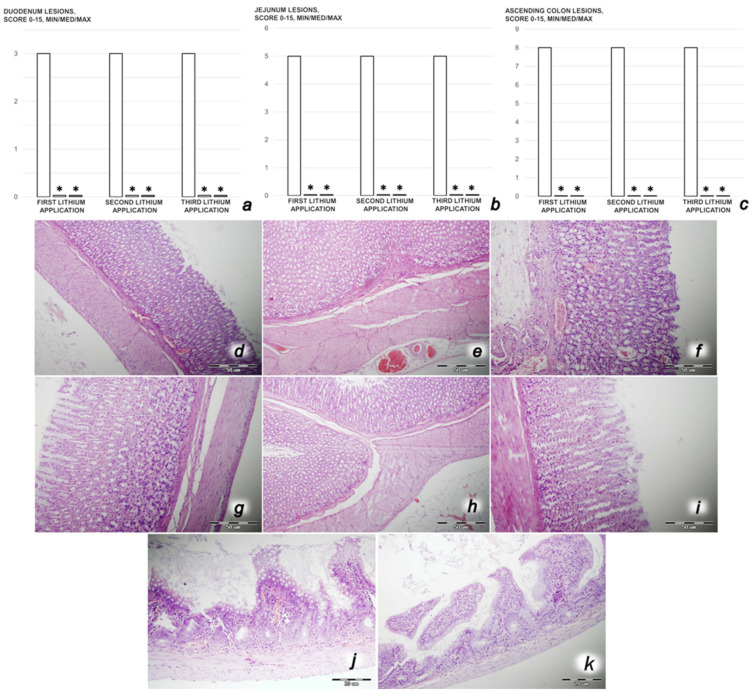
Gastrointestinal pathology. (**a**–**c**) Microscopy scoring of the lesions of the duodenum (**a**), jejunum (**b**), and ascending colon (**c**), (scored 0–15) in lithium-treated rats after the first, second, and third lithium sulphate administration (500 mg/kg/day intraperitoneally, for 3 consecutive days). BPC 157 10 µg/kg (light gray bars), 10 ng/kg (dark gray bars); saline 5 mL/kg (white bars)) given intraperitoneally after each of the lithium administrations. Min/Med/Max, six rats/group/interval. * *p* ˂ 0.05, at least, vs. control. Gastrointestinal histology. Stomach and intestine, control (**d**–**f**,**j**), BPC 157 (**g**–**i**,**k**); first lithium application (**d**,**g**), second lithium application (**e**,**h**), and third lithium application (**f**,**i**). A moderate to marked congestion of stomach wall was found in controls after each of the lithium administrations (**d**–**f**). Contrarily, lithium-treated rats that received BPC 157 regimens presented a normal gastric wall (**g**–**i**). An illustrative presentation of the intestinal mucosa marked congestion in the control lithium-treated rats after the third lithium administration (**j**) contrasting with the normal intestine mucosa presentation in lithium-treated rats that received BPC 157 regimens (**k**). (HE; magnification × 100, scale bar 50 μm (**d**–**i**); magnification × 200, scale bar 50 μm, scale bar 20 μm (**j**,**k**)).

**Figure 10 biomedicines-09-01506-f010:**
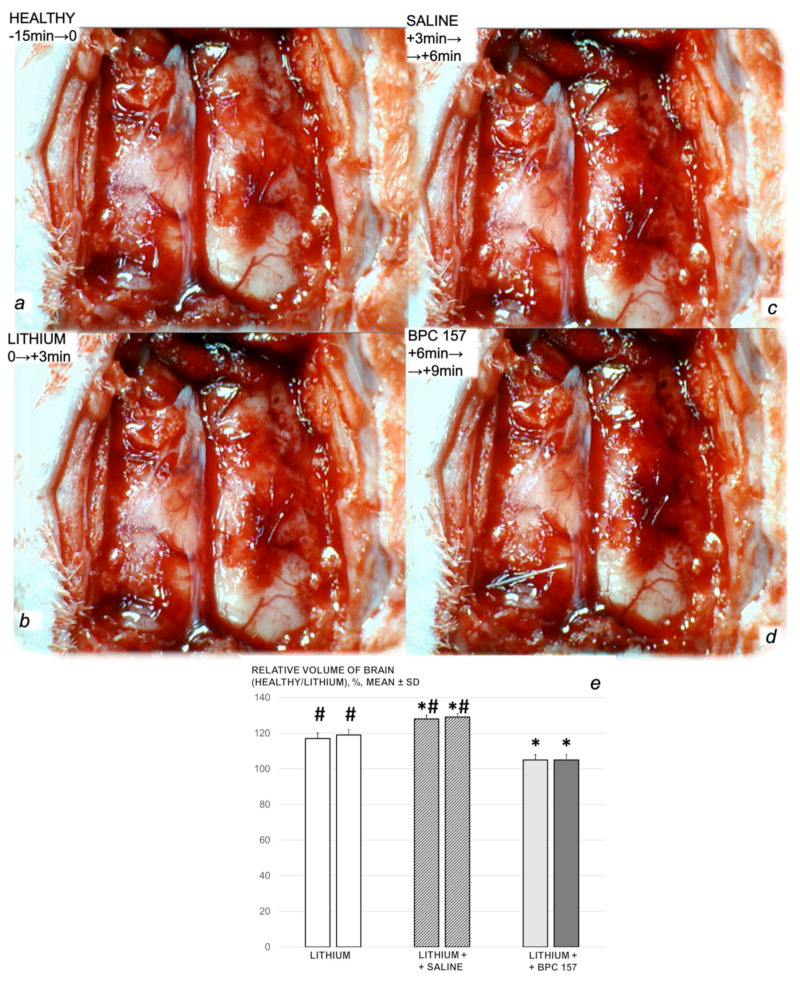
Time-line of the brain swelling and brain swelling counteraction. Brain swelling gross in vivo presentation, and lithium administration at time 0. (**a**) Before the lithium sulphate administration (time 0), presentation of the brain in the healthy rat at 15 min after calvariectomy (period −15 min→0). (**b**) Swelling progression immediately after rat received lithium administration (period 0 → +3 min). (**c**) Further swelling progression immediately after lithium-treated rats received saline therapy. (**d**) Counteracted brain swelling after lithium-treated rats received BPC 157 therapy. (**e**) Relative volume of brain swelling, volume healthy/volume after lithium, %, in the lithium-treated rats (white bars, lithium only), in the lithium-treated rats that received saline (5 mL/kg intraperitoneally (dashed bars), and finally, in the lithium-treated rats after receiving after saline, an additional application of BPC 157 10 µg/kg (light gray bars) or 10 ng/kg (dark gray bars) given intraperitoneally. Estimated periods were 3 min for the lithium administration, and subsequent administration of saline, and BPC 157. Six rats/group/interval. Means ± SD, * *p* ˂ 0.05, at least vs. control (lithium). # *p* ˂ 0.05, at least vs. healthy.

**Figure 11 biomedicines-09-01506-f011:**
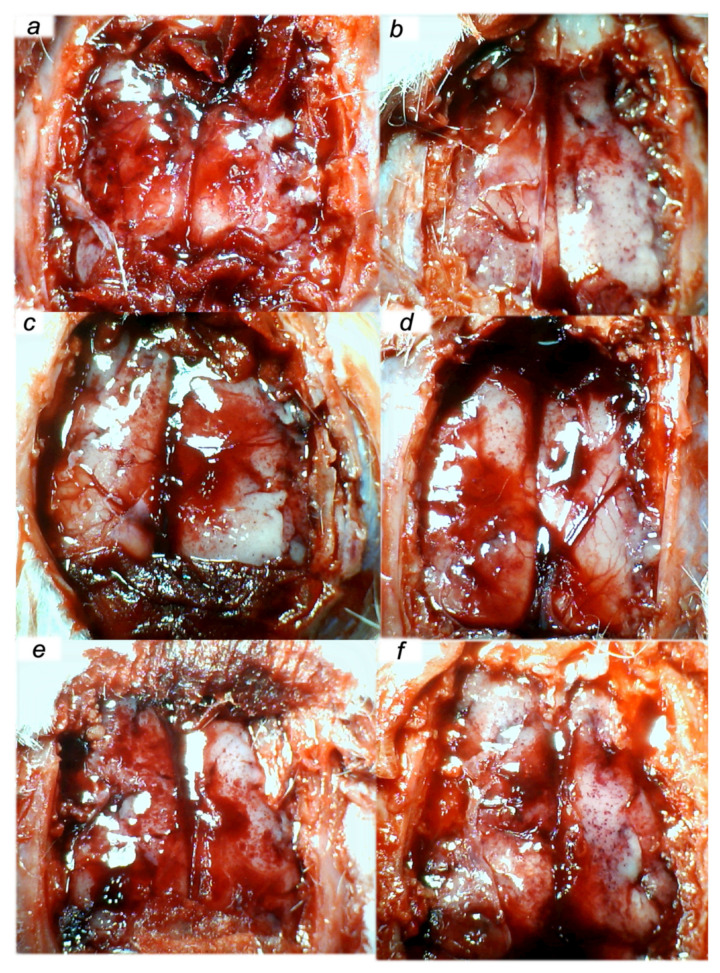
Timeline of the brain swelling and brain swelling counteraction. Brain swelling gross in vivo presentation (**a**–**f**) (first lithium application (**a**, control), (**b**, BPC 157), second lithium application (**c**, control), (**d**, BPC 157), third lithium application (**e**, control), (**f**, BPC 157)). Marked swelling was found in controls after each of the lithium administrations. Contrarily, lithium rats that received BPC 157 regimens presented counteraction of the brain swelling after each of the lithium administrations.

**Figure 12 biomedicines-09-01506-f012:**
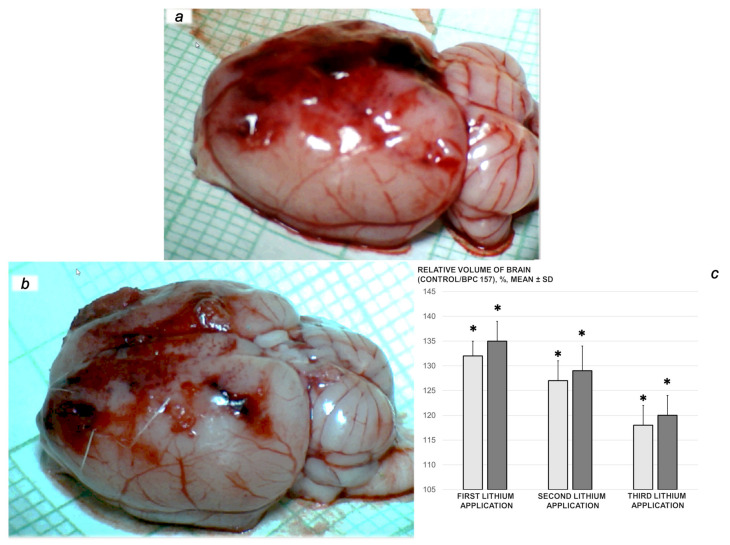
Brain gross presentation immediately after sacrifice in rats that received third lithium administration and BPC 157 (**a**) or saline (**b**) as therapy. (**c**) Relative volume of the brain, volume control/volume BPC 157, %, in the rats that received the first, second, and third lithium administration (500 mg/kg/day intraperitoneally for 3 consecutive days). BPC 157 10 µg/kg (light gray bars), 10 ng/kg (dark gray bars); saline 5 mL/kg (white bars) given intraperitoneally after each of the three lithium sulphate administrations. Six rats/group/interval. Means ± SD, * *p* ˂ 0.05, at least, vs. control.

**Figure 13 biomedicines-09-01506-f013:**
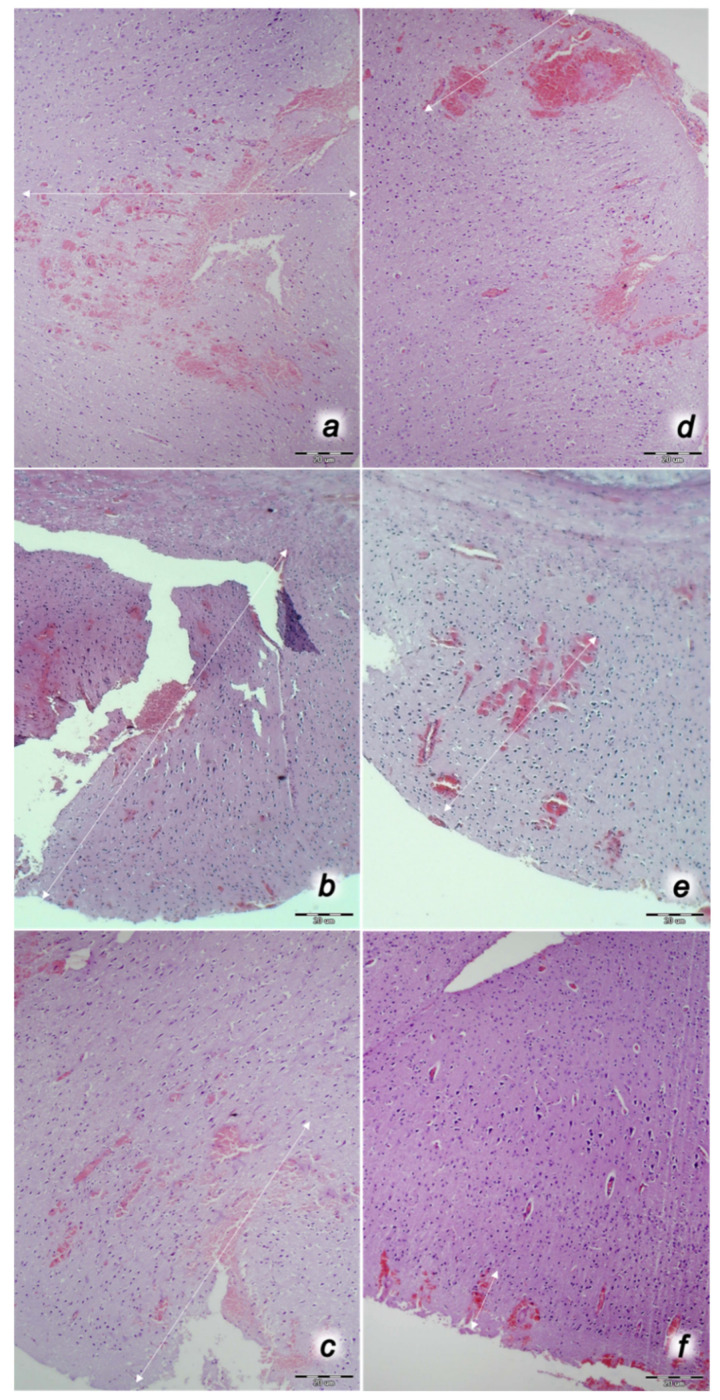
Brain pathology. Intracerebral hemorrhage in the fronto-parietal area. Control (**a**–**c**); BPC 157 (**d**–**f**); first lithium application (**a**,**d**); second lithium application (**b**,**e**); third lithium application (**c**,**f**). A marked hemorrhage was found in control lithium-treated rats after each of the lithium administrations affecting larger and deeper areas (grey and white matter). In contrast, BPC 157-treated rats had hemorrhaging located mostly in the grey matter, and smaller in diameter (lesion depth marked with double sided arrow perpendicularly orientated to the brain surface) (HE, magnification × 200) (HE, magnification × 200, scale bar 20 μm).

**Figure 14 biomedicines-09-01506-f014:**
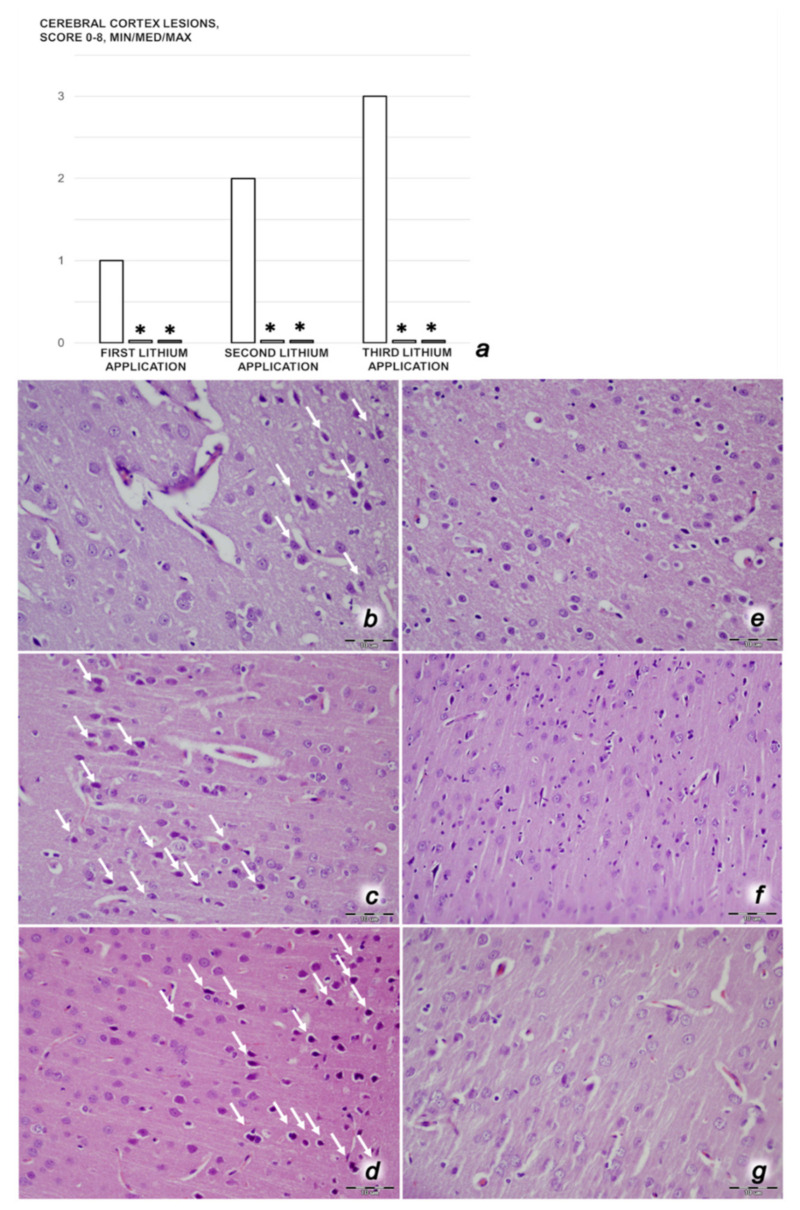
Brain pathology. Neuropathological changes of the cerebral cortex. (**a**) Semiquantitative neuropathological scoring system of cerebral cortex lesions in the lithium rats after the first, second, and third lithium administration (500 mg/kg/day intraperitoneally for 3 consecutive days). BPC 157 10 µg/kg (light gray bars), 10 ng/kg (dark gray bars); saline 5 mL/kg (white bars) given intraperitoneally after each of the three lithium administrations. Six rats/group/interval. Min/Med/Max, * *p* ˂ 0.05, at least vs. control. In control rats a continuous increase rate of affected area occurred, from grade 1 (less than 20 percent of cerebral cortex area affected with a few karyopyknotic neuronal cells) to grade 3 (up to 75 percent of cerebral cortex area affected with more extensive karyopyknotic areas) following the order of lithium administration, whereas in BPC 157 rats no neuronal changes were found. Illustrative presentation of analyzed area of cerebral cortex lesions in control (**b**–**d**), and BPC 157-treated rats (**e**,**f**); first lithium application (**b**,**e**), second lithium application (**c**,**f**), and third lithium application (**d**,**g**). A marked edema and congestion persisted in the control lithium-treated rats after each of the lithium administrations with progression from mild to marked karyopyknosis of cortical neurons (arrows). In contrast, BPC 157-treated rats showed only mild edema persisting with no neuronal changes (HE; magnification × 400, scale bar 10 μm).

**Figure 15 biomedicines-09-01506-f015:**
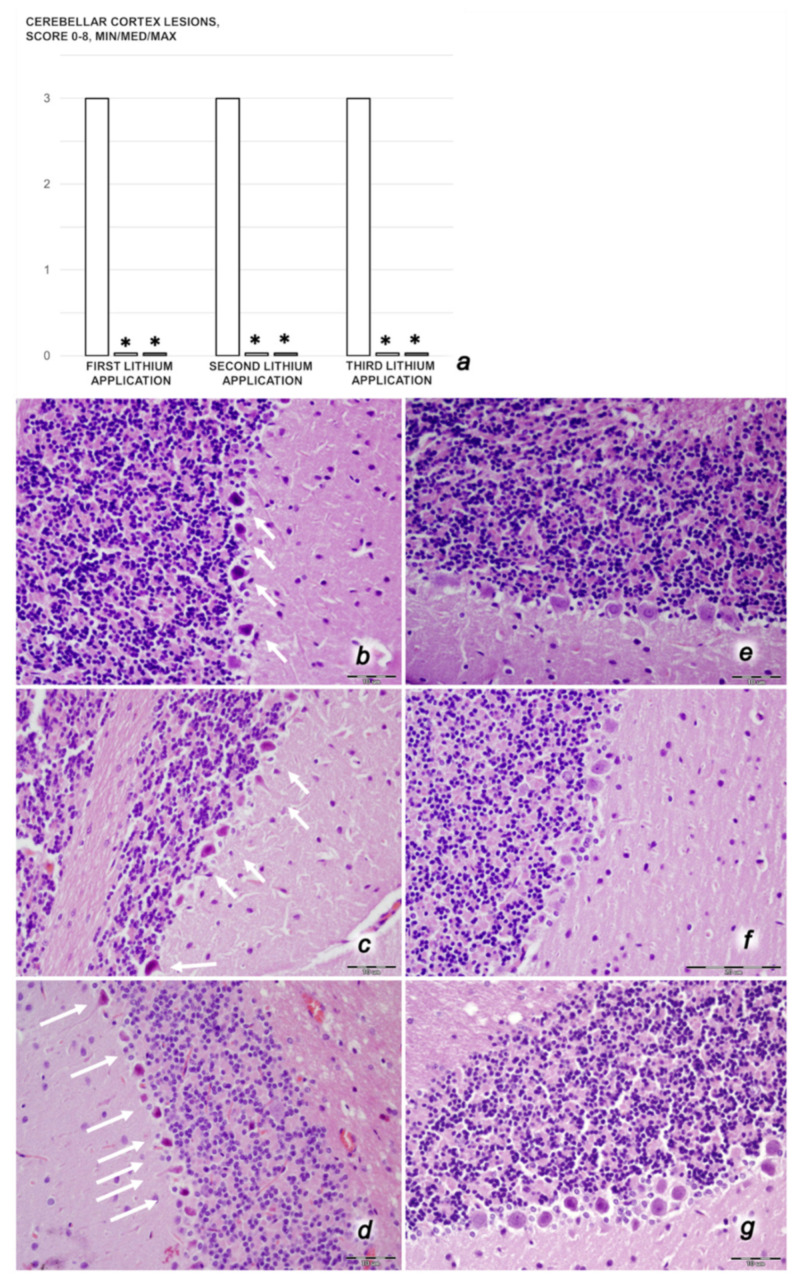
Brain pathology. Neuropathological changes of the cerebellar cortex. (**a**) Semiquantitative neuropathological scoring system analysis of lesions of the cerebellar cortex in the lithium rats after the first, second, and third lithium administration (500 mg/kg/day intraperitoneally for 3 consecutive days). BPC 157 10 µg/kg (light gray bars), 10 ng/kg (dark gray bars); saline 5 mL/kg (white bars) given intraperitoneally after each of the three lithium administrations. Six rats/group/interval. Min/Med/Max, * *p* ˂ 0.05, at least vs. control. In control rats the increased rate of cerebellar cortex area occurred to grade 3 (up to 75 percent of cerebellar cortex area affected with more extensive karyopyknotic areas) after each lithium administration, whereas in BPC 157-treated rats no neuronal changes were found. Cerebellar cortex histology (**b**–**g**). Control (**b**–**d**), BPC 157 (**e**,**f**); first lithium application (**b**,**e**), second lithium application (**c**,**f**), third lithium application (**d**,**g**). A marked karyopyknosis, degeneration (arrows), and loss of Purkinje cells of the cerebellar cortex, persistent congestion, and cerebellar edema presented in the control lithium-treated rats after each of the lithium administrations. In contrast, BPC 157-treated rats showed a normal structure of the cerebellar cortex (HE; magnification × 400, scale bar 10 μm).

**Figure 16 biomedicines-09-01506-f016:**
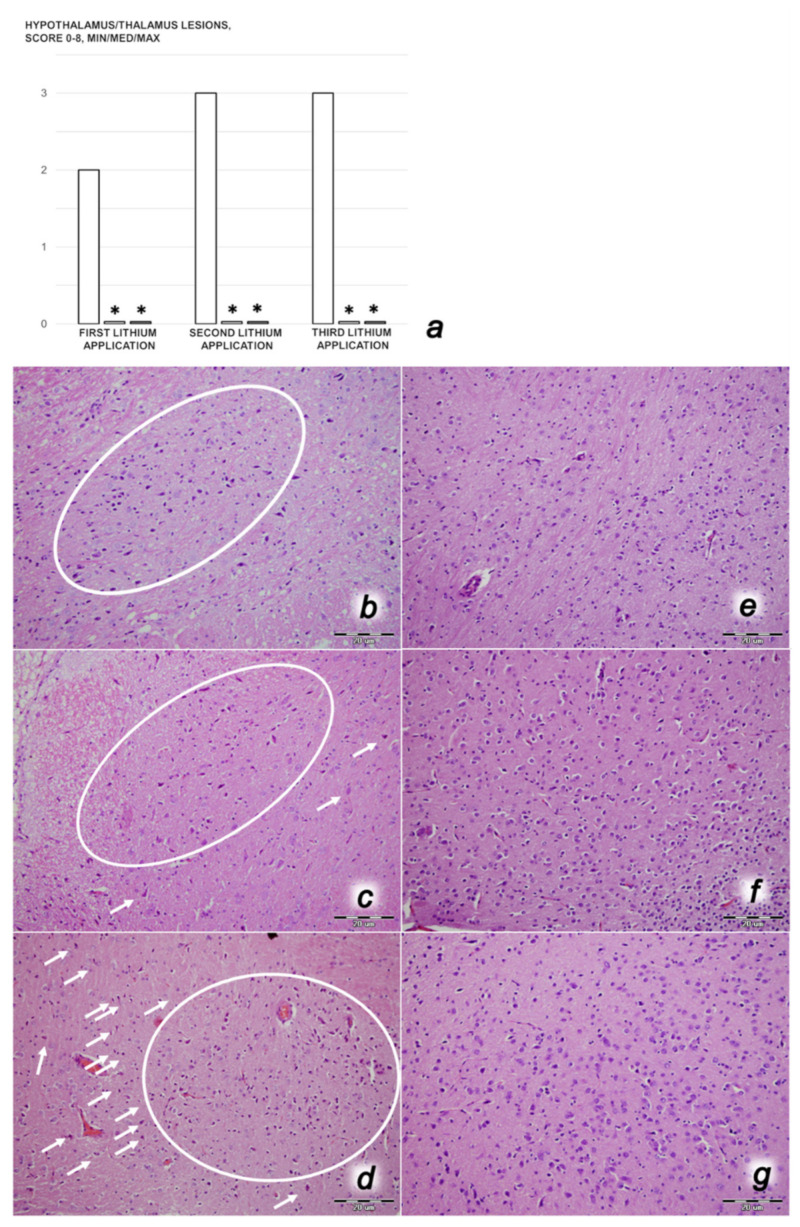
Brain pathology. Neuropathological changes of the hypothalamus. (**a**) Semiquantitative neuropathological scoring system of the lesions of the hypothalamus in the lithium-treated rats after the first, second, and third lithium administration (500 mg/kg/day intraperitoneally for 3 consecutive days). BPC 157 10 µg/kg (light gray bars), 10 ng/kg (dark gray bars); saline 5 mL/kg (white bars) given intraperitoneally after each of the three lithium administrations. Six rats/group/interval. Min/Med/Max, * *p* ˂ 0.05, at least vs. control. In control rats, a continuous increase in the rate of affected area occurred, from grade 2 (up to 50 percent of hypothalamus cortex area affected with a patchy karyopyknotic areas) after the first lithium administration, to grade 3 (up to 75 percent of hypothalamus cortex area affected with more extensive of) after the second and third lithium administrations, whereas in BPC 157-treated rats no neuronal changes were found. Hypothalamus histology (**b**–**g**). Control (**b**,**d**,**f**), BPC 157 (**c**,**e**,**g**); first lithium application (**b**,**c**), second lithium application (**d**,**e**), and third lithium application (**f**,**g**). A marked karyopyknosis of hypothalamic neurons (circles and arrows) presented in the control lithium-treated rats after each of the lithium administrations. In contrast, BPC 157-treated rats showed a normal structure of the hypothalamus (HE; magnification × 200, scale bar 20 μm).

**Figure 17 biomedicines-09-01506-f017:**
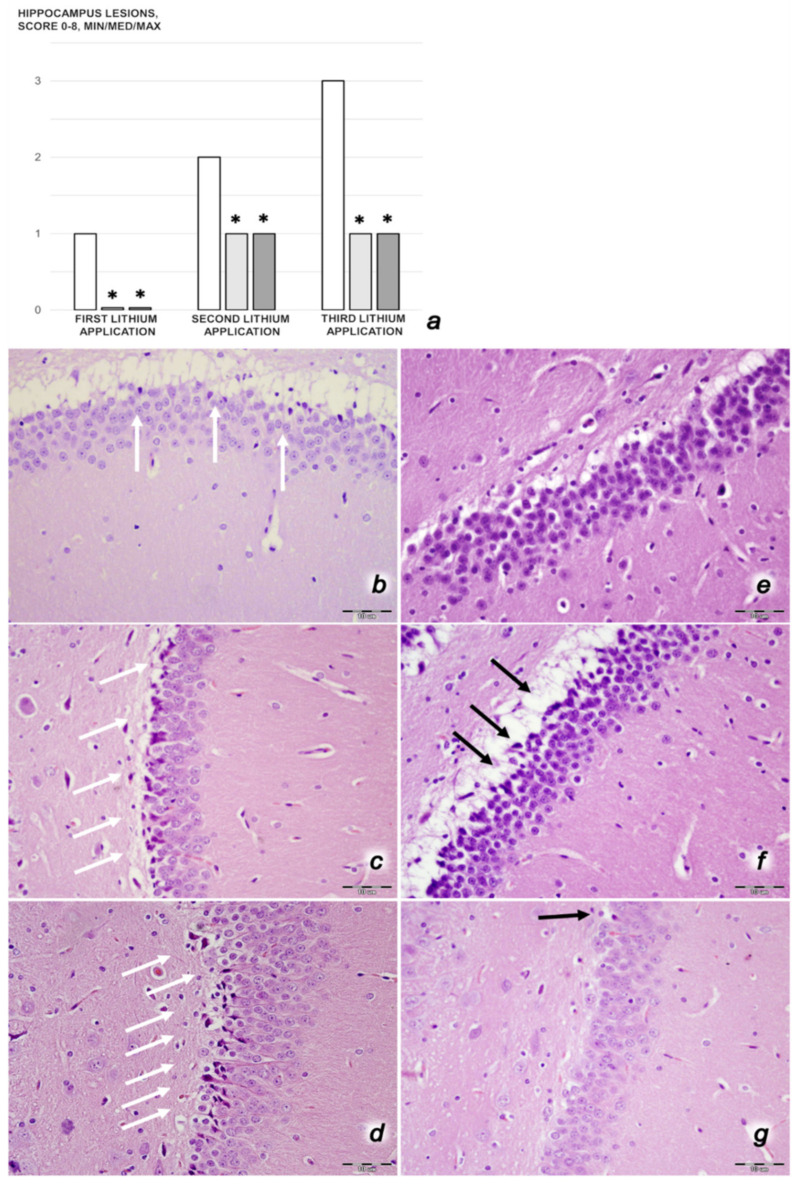
Brain pathology. Neuropathological changes of the hippocampus. (**a**) Semiquantitative microscopy scoring of the hippocampal lesions in the lithium-treated rats after the first, second, and third lithium administration (500 mg/kg/day intraperitoneally for 3 consecutive days). BPC 157 10 µg/kg (light gray bars), 10 ng/kg (dark gray bars); saline 5 mL/kg (white bars) given intraperitoneally after each of the three lithium administrations. Six rats/group/interval. Min/Med/Max, * *p* ˂ 0.05, at least vs. control. In control rats a continuous increase in the rate of affected area occurred, from grade 1 (less than 20 percent of hippocampus cortex area affected with a few karyopyknotic neuronal cells), to grade 3 (up to 75 percent of hippocampus cortex area affected with more extensive karyopyknotic areas) following the order of lithium administration, whereas in BPC 157-treated rats grade 1 neuronal changes occurred after the second and third lithium administrations, with no changes after the first lithium administration. Illustrative hippocampus histology (**b**–**g**). Control (**b**–**d**), BPC 157 (**e**,**f**); first lithium application (**b**,**e**), second lithium application (**c**,**f**), and third lithium application (**d**,**g**). A progressive karyopyknosis of pyramidal cells of the hippocampus presented in the control lithium-treated rats after each of the lithium administrations, and particularly after the third lithium administration (white arrows). In contrast, BPC 157-treated rats showed only a few, scattered karyopyknotic cells after the second and third lithium administrations (black arrows). (HE; magnification × 400, scale bar 10 μm).

**Figure 18 biomedicines-09-01506-f018:**
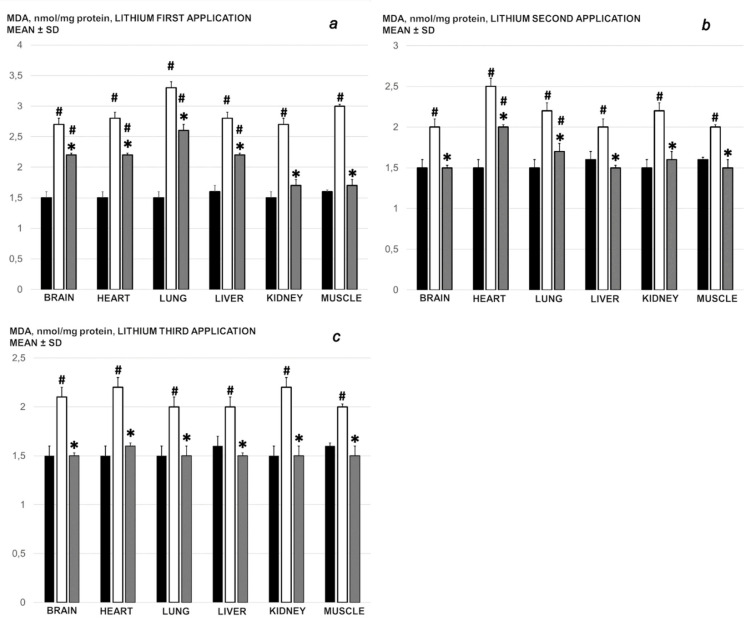
Oxidative stress (MDA, nmol/g protein), in healthy rats (black bars) and in the lithium-treated rats after the first (**a**), second (**b**), and third (**c**) lithium administration (500 mg/kg/day intraperitoneally for 3 consecutive days). BPC 157 10 ng/kg (dark gray bars) or saline 5 mL/kg (white bars) given intraperitoneally after each of the three lithium administrations. Six rats/group/interval. Means ± SD, # *p* ˂ 0.05, at least vs. healthy; * *p* ˂ 0.05, at least vs. control.

**Figure 19 biomedicines-09-01506-f019:**
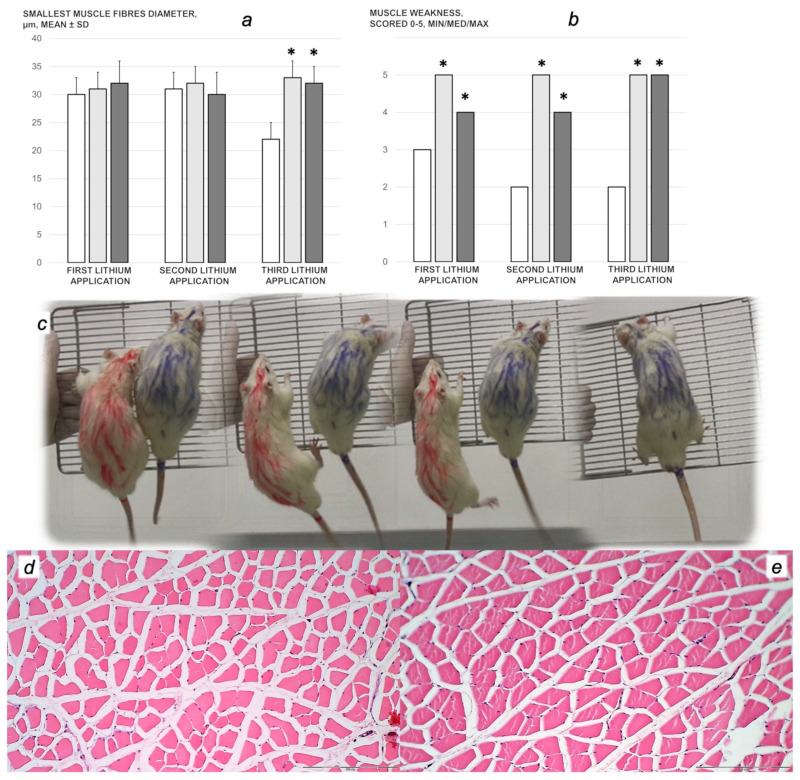
Muscle fiber diameter (µm, means ± SD) (**a**) and muscle weakness (scored 0–5, Min/Med/Max) (**b**) in the lithium-treated rats after the first, second, and third lithium administration (500 mg/kg/day intraperitoneally for 3 consecutive days). BPC 157 10 ng/kg (dark gray bars) or saline 5 mL/kg (white bars) given intraperitoneally after each of the three lithium administrations. Six rats/group/interval. * *p* ˂ 0.05, at least vs. control. (**c**) Illustrative presentation of muscle weakness in rats after first lithium administration (red (control), blue (BPC 157)). Illustrative histology presentation of quadriceps muscle after third lithium administration, presentation with a smaller diameter of muscle fibers in control rats (**d**) and preserved muscle presentation in BPC 157-treated rats (**e**) (HE, × 100; scale bar 200 μm).

**Figure 20 biomedicines-09-01506-f020:**
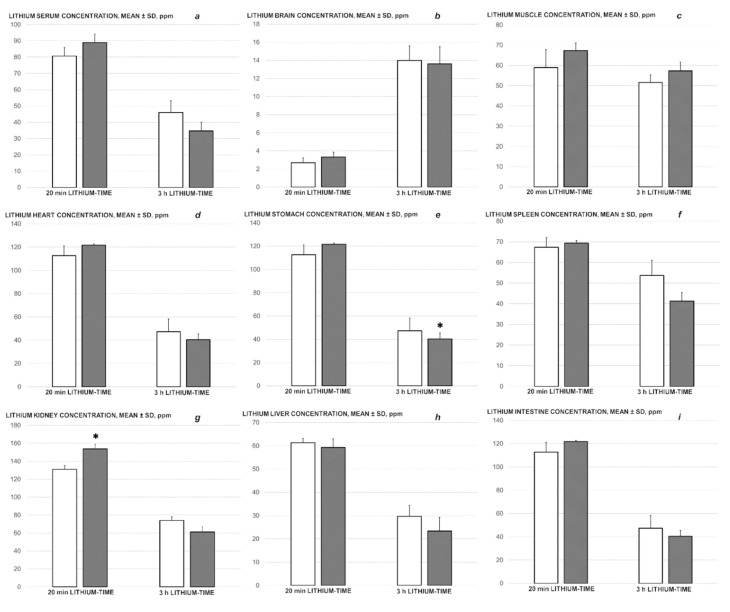
At 20 min or 3 h after administration of lithium sulfate (500 mg/kg ip) (lithium-time), followed by BPC 157 10 ng/kg ip (dark gray bars), or saline 5 mL/kg ip (white bars); concentrations of lithium were assessed in samples of rat serum (**a**), brain (**b**), muscle (**c**), heart (**d**), stomach (**e**), spleen (**f**), kidney (**g**), liver (**h**), and intestine (**i**). Means ± SD, * *p* ˂ 0.05, at least vs. control.

**Table 1 biomedicines-09-01506-t001:** The neuropathological scores.

Brain Area	Grading	Percent Area Affected	Morphological Changes
Cerebral and cerebellar cortex, hypothalamus, thalamus, hippocampus	1	≤10	Small, patchy, complete, or incomplete infarcts
	2	20–30	Partly confluent complete or incomplete infarcts
	3	40–60	Large confluent compete infarcts
	4	>75	In cortex; total disintegration of the tissue, in hypothalamus, thalamus, hippocampus; large complete infarcts
Cerebral and cerebellar cortex, hypothalamus, thalamus, hippocampus	1	≤20	A few karyopyknotic of neuronal cells
	2	50	Patchy areas of karyopyknotic areas
	3	75	More extensive of karyopyknotic areas
	4	100	Complete infarction

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
