# Peer review of "Over-Dose Lithium Toxicity as an Occlusive-like Syndrome in Rats and Gastric Pentadecapeptide BPC 157"

_biomedicines, 2021, doi:10.3390/biomedicines9111506_

Round 1

Reviewer 1 Report

The manuscript entitled " Over-dose lithium toxicity as an occlusive-like syndrome in 2 rats and gastric pentadecapeptide BPC 157" by Strbe et al., focused on the effect of the over-dose lithium in rats as an occlusive-like syndrome, due to endothelium impairment.

I have some comments:

- A careful stylistic and linguistic revision is required.

-In the majority of the figures is just explained in the figure legend the bars what represent. It may be better for the reader to add that directly in the figure.

-Please, add scale bar to heart, lung, liver, kidney, gastrointestinal, brain pathology

-Please, add statical analysis in figure legends

Author Response

Dear Editor,

Re: biomedicines-1358823 

Hope you will not mind this delay providing that all of the requests of the reviewers were accordingly resolved, and that manuscript appears to be adequately improved, likely more suited for final presentation.

You should specifically indicate that along with the language correction done by the professional office,  the newest evidence that in meantime appeared in Biomedicines (Knezevic, M.; Gojkovic, S.; Krezic, I.; Zizek, H.; Malekinusic, D.; Vrdoljak, B.; Vranes, H.; Knezevic, T.; Barisic, I.; Horvat Pavlov, K.; et al. Occlusion of the superior mesenteric artery in rats reversed by collateral pathways activation: Gastric pentadecapeptide BPC 157 therapy counteracts multiple organ dysfunction syndrome; intracranial, portal and caval hypertension; and aortal hypotension. Biomedicines 2021, 9, 609; Gojkovic, S.; Krezic, I.; Vrdoljak, B.; Malekinusic, D.; Barisic, I.; Petrovic, A.; Horvat Pavlov, K.; Kolovrat, M.; Duzel, A.; Knezevic, M.; et al. Pentadecapeptide BPC 157 resolves suprahepatic occlusion of the inferior caval vein, Budd-Chiari syndrome model in rats. World J. Gastrointest. Pathophysiol. 2020, 11, 1–19. ; Knezevic, M.; Gojkovic, S.; Krezic, I.; Zizek, H.; Vranes, H.; Malekinusic, D.; Vrdoljak, B.; Knezevic, T.; Pavlov, K.H.; Drmic, D.; et al. Complex syndrome of the complete occlusion of the end of the superior mesenteric vein, opposed with the stable gastric pentadecapeptide BPC 157 in rats. Biomedicines 2021, 9, 1029. Knezevic, M.; Gojkovic, S.; Krezic, I.; Zizek, H.; Malekinusic, D.; Vrdoljak, B.; Knezevic, T.; Vranes, H.; Drmic, D.; Staroveski, M.; et al. Occluded superior mesenteric artery and vein. Therapy with the stable gastric pentadecapeptide BPC 157. Biomedicines 2021, 9, 609. Gojkovic, S.; Krezic, I.; Vranes, H.; Zizek, H.; Drmic, D.; Pavlov, K.H.; Petrovic, A.; Batelja, L.; Milavic, M.; Sikiric, S.; et al. BPC 157 therapy and the permanent occlusion of the superior sagittal sinus in rat. Vascular recruitment. Biomedicines 2021, 9, 744.) is included in the revised manuscript to support the original presentation of the lithium toxicity as an „occlusion-like“ syndrome, and consequent beneficial effect of the pentadecapeptide BPC 157 therapy.

The following comments were given by the reviewers.

Reviewer 1

Open Review

English language and style

( ) Extensive editing of English language and style required
(x) Moderate English changes required
( ) English language and style are fine/minor spell check required
( ) I don't feel qualified to judge about the English language and style

Yes

Can be improved

Must be improved

Not applicable

Does the introduction provide sufficient background and include all relevant references?

( )

(x)

( )

( )

Is the research design appropriate?

( )

(x)

( )

( )

Are the methods adequately described?

( )

(x)

( )

( )

Are the results clearly presented?

( )

(x)

( )

( )

Are the conclusions supported by the results?

( )

(x)

( )

( )

Comments and Suggestions for Authors

The manuscript entitled " Over-dose lithium toxicity as an occlusive-like syndrome in 2 rats and gastric pentadecapeptide BPC 157" by Strbe et al., focused on the effect of the over-dose lithium in rats as an occlusive-like syndrome, due to endothelium impairment.

I have some comments:

- A careful stylistic and linguistic revision is required.

-In the majority of the figures is just explained in the figure legend the bars what represent. It may be better for the reader to add that directly in the figure.

-Please, add scale bar to heart, lung, liver, kidney, gastrointestinal, brain pathology

-Please, add statical analysis in figure legend

To the  comments of  the reviewers see our reply and arguments.

Reviewer 1

The manuscript entitled " Over-dose lithium toxicity as an occlusive-like syndrome in 2 rats and gastric pentadecapeptide BPC 157" by Strbe et al., focused on the effect of the over-dose lithium in rats as an occlusive-like syndrome, due to endothelium impairment.

I have some comments:

- A careful stylistic and linguistic revision is required.

-In the majority of the figures is just explained in the figure legend the bars what represent. It may be better for the reader to add that directly in the figure.

-Please, add scale bar to heart, lung, liver, kidney, gastrointestinal, brain pathology

-Please, add statical analysis in figure legend

Ad Reviewer 1. Acknowledged. As mentioned above, new supportive evidence, very recently presented in Biomedicines, were included in the revised version, and thereby, the whole manuscript is accordingly checked by the professional language office, and hopefully markedly improved. The whole text and Figures are modified as requested by the reviewer.

Summarizing, we hope that the   comments of  the reviewers are adequately resolved, and that the manuscript is now more suited for final presentation.

Looking forward to hearing from you very soon

Sincerely

Predrag Sikiric, MD, PhD

Professor

Reviewer 2 Report

The authors investigated whether stable gastric pentadecapeptide BPC 157 had a protective effect on the lithium-induced occlusive-like syndrome. The result showed that BPC 157 can counteract brain swelling, hypertension, weakness, and multiorgan damage caused by the acute super high dose of lithium. However, there is some defect cannot be bypassed in this article. Also, the manuscript was poorly prepared to make it unreadable.

  1. Although the author mentioned the dose selection in the article, they still didn’t clarify the reason they chose so high a dose. In the clinic, lithium usually is given orally. Meanwhile, the blood concentration will be monitored during the regimen. Here, the author should provide the blood concentration of lithium. Otherwise, they cannot exclude that the relatively good results in the treatment group are from the low drug concentration in the blood.
  2. Usually, the toxicity of lithium in the clinic result from a high blood concentration of lithium after taking the drug several days later. However, all the experiments in the article are performed in the short term. Can the BPC 157 continuously protect an animal from toxicity for a long time?
  3. The captions of Figure 14 and Figure 15 in the article are similar. The author should clarify the corresponding scoring system for each figure.

4. In the oxidative stress study, the level of MDA in the tissue decreased with the administration of lithium. However, the tissue damage increased with the administration of lithium. Why? In addition, a normal level of MDA in the tissue is missing.

Author Response

Dear Editor,

Re: biomedicines-1358823 

Hope you will not mind this delay providing that all of the requests of the reviewers were accordingly resolved, and that manuscript appears to be adequately improved, likely more suited for final presentation.

You should specifically indicate that along with the language correction done by the professional office,  the newest evidence that in meantime appeared in Biomedicines (Knezevic, M.; Gojkovic, S.; Krezic, I.; Zizek, H.; Malekinusic, D.; Vrdoljak, B.; Vranes, H.; Knezevic, T.; Barisic, I.; Horvat Pavlov, K.; et al. Occlusion of the superior mesenteric artery in rats reversed by collateral pathways activation: Gastric pentadecapeptide BPC 157 therapy counteracts multiple organ dysfunction syndrome; intracranial, portal and caval hypertension; and aortal hypotension. Biomedicines 2021, 9, 609; Gojkovic, S.; Krezic, I.; Vrdoljak, B.; Malekinusic, D.; Barisic, I.; Petrovic, A.; Horvat Pavlov, K.; Kolovrat, M.; Duzel, A.; Knezevic, M.; et al. Pentadecapeptide BPC 157 resolves suprahepatic occlusion of the inferior caval vein, Budd-Chiari syndrome model in rats. World J. Gastrointest. Pathophysiol. 2020, 11, 1–19. ; Knezevic, M.; Gojkovic, S.; Krezic, I.; Zizek, H.; Vranes, H.; Malekinusic, D.; Vrdoljak, B.; Knezevic, T.; Pavlov, K.H.; Drmic, D.; et al. Complex syndrome of the complete occlusion of the end of the superior mesenteric vein, opposed with the stable gastric pentadecapeptide BPC 157 in rats. Biomedicines 2021, 9, 1029. Knezevic, M.; Gojkovic, S.; Krezic, I.; Zizek, H.; Malekinusic, D.; Vrdoljak, B.; Knezevic, T.; Vranes, H.; Drmic, D.; Staroveski, M.; et al. Occluded superior mesenteric artery and vein. Therapy with the stable gastric pentadecapeptide BPC 157. Biomedicines 2021, 9, 609. Gojkovic, S.; Krezic, I.; Vranes, H.; Zizek, H.; Drmic, D.; Pavlov, K.H.; Petrovic, A.; Batelja, L.; Milavic, M.; Sikiric, S.; et al. BPC 157 therapy and the permanent occlusion of the superior sagittal sinus in rat. Vascular recruitment. Biomedicines 2021, 9, 744.) is included in the revised manuscript to support the original presentation of the lithium toxicity as an „occlusion-like“ syndrome, and consequent beneficial effect of the pentadecapeptide BPC 157 therapy.

The following comments were given by the reviewers.

Reviewer 2

Open Review

English language and style

(x) Extensive editing of English language and style required
( ) Moderate English changes required
( ) English language and style are fine/minor spell check required
( ) I don't feel qualified to judge about the English language and style

Yes

Can be improved

Must be improved

Not applicable

Does the introduction provide sufficient background and include all relevant references?

( )

(x)

( )

( )

Is the research design appropriate?

( )

( )

(x)

( )

Are the methods adequately described?

(x)

( )

( )

( )

Are the results clearly presented?

( )

(x)

( )

( )

Are the conclusions supported by the results?

( )

( )

(x)

( )

Comments and Suggestions for Authors

The authors investigated whether stable gastric pentadecapeptide BPC 157 had a protective effect on the lithium-induced occlusive-like syndrome. The result showed that BPC 157 can counteract brain swelling, hypertension, weakness, and multiorgan damage caused by the acute super high dose of lithium. However, there is some defect cannot be bypassed in this article. Also, the manuscript was poorly prepared to make it unreadable.

  1. Although the author mentioned the dose selection in the article, they still didn’t clarify the reason they chose so high a dose. In the clinic, lithium usually is given orally. Meanwhile, the blood concentration will be monitored during the regimen. Here, the author should provide the blood concentration of lithium. Otherwise, they cannot exclude that the relatively good results in the treatment group are from the low drug concentration in the blood.
  2. Usually, the toxicity of lithium in the clinic result from a high blood concentration of lithium after taking the drug several days later. However, all the experiments in the article are performed in the short term. Can the BPC 157 continuously protect an animal from toxicity for a long time?
  3. The captions of Figure 14 and Figure 15 in the article are similar. The author should clarify the corresponding scoring system for each figure.
  1. In the oxidative stress study, the level of MDA in the tissue decreased with the administration of lithium. However, the tissue damage increased with the administration of lithium. Why? In addition, a normal level of MDA in the tissue is missing.

To the  comments of  the reviewers see our reply and arguments.

The authors investigated whether stable gastric pentadecapeptide BPC 157 had a protective effect on the lithium-induced occlusive-like syndrome. The result showed that BPC 157 can counteract brain swelling, hypertension, weakness, and multiorgan damage caused by the acute super high dose of lithium. However, there is some defect cannot be bypassed in this article. Also, the manuscript was poorly prepared to make it unreadable.

  1. Although the author mentioned the dose selection in the article, they still didn’t clarify the reason they chose so high a dose. In the clinic, lithium usually is given orally. Meanwhile, the blood concentration will be monitored during the regimen. Here, the author should provide the blood concentration of lithium. Otherwise, they cannot exclude that the relatively good results in the treatment group are from the low drug concentration in the blood.
  2. Usually, the toxicity of lithium in the clinic result from a high blood concentration of lithium after taking the drug several days later. However, all the experiments in the article are performed in the short term. Can the BPC 157 continuously protect an animal from toxicity for a long time?

Ad Reviewer 2. Acknowledged. To reply to the comment of the reviewer see additional assessment (Method, section 2.3., Results, section 3.6.,Figure 20), and lithium concentration, not only in the serum, but also in corresponding organs as well. The obtained evidence excludes the possibility mentioned by the reviewer (see also Discussion, paragraph 3):

Of note, at that time, BPC 157 rats share the same high serum lithium concentration. Besides, these BPC 157 effects may reveal an intriguing point in relation to high serum lithium level-normal functioning, instead of high serum lithium-disturbed function as may be expected [17]. This strong beneficial effect of BPC 157 can essentially replicate a strong innate counteracting potential, with a potential life-saving effect, noted also with constant severe and eventually lethal hyperkalemia (counteracted were lethal effect, severe arrhythmias, and muscle disability while serum potassium concentration remained high) after intraperitoneal potassium chloride over-dose challenge [52]. Note, considering lithium-endothelium effect [1], potassium is also shown to cause endothelial damage [9]. Possibly, the high serum (lithium and potassium) values would be suggestive for an immediate vascular failure (and then, continuous), and consequent tissue damaging effect – that may be counteracted, immediately and continuously, by BPC 157 therapy.  On the other hand, evidently, such lithium over-load (high serum levels, and low brain levels [17]), excludes that the relatively good results in the treatment group are from the lower drug concentration in the blood. Finally, in patients,  aversion to lithium can be elicited even when serum lithium levels are within the “therapeutic” range [17,67].

To the specific argument, raised in the point 2, the high toxicity of the lithium in the control rats precludes much longer experiments, and thereby the evidence that BPC 157 may be effective continuously is based on the following notation.

Possibly, the high serum (lithium and potassium) values would be suggestive for an immediate vascular failure (and then, continuous), and consequent tissue damaging effect – that may be counteracted, immediately and continuously, by BPC 157 therapy. 

   3. The captions of Figure 14 and Figure 15 in the article are similar. The author should clarify the corresponding scoring system for each figure.

Ad Reviewer 2. Acknowledged.

  1. In the oxidative stress study, the level of MDA in the tissue decreased with the administration of lithium. However, the tissue damage increased with the administration of lithium. Why? In addition, a normal level of MDA in the tissue is missing.

Ad Reviewer 2. Acknowledged. This point is now better explained. Considering the presented results, the lithium oxidative stress is along with the noted “occlusive-like” syndrome, and severe tissue damages that were induced. This point is emphasized in Discussion, paragraph 7.

In all the tissues tested, the BPC 157’s actions against the damaging effects to blood vessel function can be also combined with an anti-oxidant effect in the lithium-treated rats covered with BPC 157 who had reduced MDA values, as a confirmative result of both preserved and rescued tissue integrity and vein integrity [2-7]. As previously described, this free radical scavenger effect occurs in both ischemic and reperfusion conditions in the various tissues (i.e., brain, colon, duodenum, cecum, liver and veins) and plasma [2-7,21-24,39-41].

General point, considering the possible beneficial effect of the lithium, including the effect on the oxidative stress, is mentioned in the concluding paragraph. Two additional references (for both beneficial effect and damaging effect) were included. 

Furthermore, and the BPC 157 and lithium issue elaborated in this study needs additional studies, namely, considering the opposite beneficial effects of lithium known in the endothelium and each of the mentioned organs [117-123], and the evidence that lithium has been found to prevent and/or reverse DNA damage, free-radical formation and lipid peroxidation in diverse models [124] as well as induce oxidative damage and inflammation [125]. However, we have revealed the toxicity of the lithium complex as an occlusive-like model, sharing pathology with syndromes after major vessel occlusion [2-8] or intragastric absolute alcohol administration [9]; intracranial, portal and caval hypertension; aortal hypotension; and multiorgan dysfunction syndrome, and oxidative stress as well as BPC 157 functioning and therapy counteracting high lithium dose intoxication in rats.

Summarizing, we hope that the   comments of  the reviewers are adequately resolved, and that the manuscript is now more suited for final presentation.

Looking forward to hearing from you very soon

Sincerely

Predrag Sikiric, MD, PhD

Professor

Round 2

Reviewer 2 Report

  1. The captions of Figure 14 and Figure 15 in the article have not been clarified. In the caption of Figure 3, the authors said white bars are control groups, but there is no white bar in the figure.
  2. The captions of Figure 20 didn't describe which experimental group the bars represent.
  3. The authors stated that concentrations of lithium in samples of rat serum evidenced the highest values in controls as well as BPC 157 treated rats. However, they only provided the concentration of lithium at 20min. How can they make this conclusion?
  4. Please clarify the time you collect samples for lithium analysis. The authors said the sample is collected 20 mins after administration. is it 20mins after administration of lithium or BPC157? 
  5. 20mins after administration is not a good time point for a drug distribution study when most of their experiments were conducted at 3hours after administration.  Also, if the authors cannot provide drug distribution at different time points, at least, the drug concentration in blood at different time points after administration of BPC157 should be provided to demonstrate the administration BPC157 didn't change the concentration of lithium in the blood. In addition, since the concentration of lithium in the blood is such high, perfusion is necessary when authors doing drug distribution studies.

Author Response

Comments and Suggestions for Authors

1. The captions of Figure 14 and Figure 15 in the article have not been clarified. In the caption of Figure 3, the authors said white bars are control groups, but there is no white bar in the figure.

Acknowledged and corrected.

 2. The captions of Figure 20 didn't describe which experimental group the bars represent

Acknowledged and corrected.

3. The authors stated that concentrations of lithium in samples of rat serum evidenced the highest values in controls as well as BPC 157 treated rats. However, they only provided the concentration of lithium at 20min. How can they make this conclusion?

Acknowledged and clarified. In the previous version section 3.6. it was stated At the time, when strict difference is already present between the regular downhill injury course in lithium overloaded rats, and beneficial effect of the BPC 157 therapy (see above), BPC 157-lithium-treated rats and saline-treated lithium-rats share the same highest serum lithium concentration.

Thus, it is clear that this particular combination (the similar lithium concentrations in the serum in all lithium-overloaded rats along with the evidently attenuated course in BPC 157 treated rats) may be a strong argument that the therapy effect is not related to the possible effect on lithium serum concentrations. This point is further elaborated in Discussion, line 755-771.

In the revised manuscript (in particular in consideration of the comparable findings at 3 h) this point is further elaborated in both Results (3.6) and Discussion (line 777-800).

At the early  time (i.e., 20 min lithium-time), the early lesions presentation in the lithium overloaded rats  present a strict difference,  the regular downhill injury course in the control-lithium rats, and markedly attenuated course in the lithium-BPC 157-rats  due to the beneficial effect of the BPC 157 therapy (see above, i.e., Figure 10 (brain swelling), Figure 19 (muscle weakness)). However,  BPC 157-lithium-treated rats and saline-treated lithium-rats share the same high serum lithium concentration. Likewise,  concentrations of lithium in samples of rat organs showed quite similar values in the brain, heart,  liver, spleen, muscle, stomach, and intestine. The concentrations of lithium in the samples of kidney were higher in the BPC 157-treated lithium-rats (Figure 20). An alike situation appeared at  at the later  time (i.e., 3 h lithium-time). There was an even advanced lithium course (lower concentration in the serum, higher in the brain), obviously   different  lesions severity in the lithium-overloaded rats (saline-rats  vs. BPC 157-rats), but all of them have merely similar lithium concentrations in the serum.  Similar  lithium concentrations were noted in the organs as well (except to the stomach, lower concentration in the lithium BPC 157-treated rats)  (Figure 20).

Of note, at that early time (i.e., 20 min lithium-time), lithium-overloaded BPC 157 rats with apparently attenuated lithium-course share the same high serum lithium concentration as the lithium-overloaded control-downhill injury course (for illustration, see Figure 10  (brain swelling, and BPC 157 effect), and Figure 19 (muscle weakness and BPC 157 effect), likely indicative for further either  downhill course (lithium-overloaded control rats)  or attenuated lithium-course (lithium-overloaded BPC 157 rats)). An alike situation appeared with the advanced lithium course (at 3 h lithium-time; lesser lithium concentration in the serum, higher in the brain [17]). Besides, these BPC 157 effects may reveal an intriguing point in relation to high serum lithium level-normal functioning, instead of high serum lithium-disturbed function as may be expected [17]. This strong beneficial effect of BPC 157 can essentially replicate a strong innate counteracting potential, with a potential life-saving effect, noted also with constant severe and eventually lethal hyperkalemia (counteracted were lethal effect, severe arrhythmias, and muscle disability while serum potassium concentration remained high) after intraperitoneal potassium chloride over-dose challenge [52]. Note, considering lithium-endothelium effect [1], potassium is also shown to cause endothelial damage [9]. Possibly, the high serum (lithium and potassium) values would be suggestive for an immediate vascular failure (and then, continuous), and consequent tissue damaging effect – that may be counteracted, immediately and continuously, by BPC 157 therapy.  On the other hand, evidently, such lithium over-load (the same high serum levels, and low brain levels at early interval, and the same low serum levels, and high  brain levels at late interval [17]), excludes that the relatively good results in the treatment group are from the lower drug concentration in the blood. Finally, in patients,  aversion to lithium can be elicited even when serum lithium levels are within the “therapeutic” range [17,67].

4. Please clarify the time you collect samples for lithium analysis. The authors said the sample is collected 20 mins after administration. is it 20mins after administration of lithium or BPC157? 

Acknowledged and clarified (lithium-time).

5. 20mins after administration is not a good time point for a drug distribution study when most of their experiments were conducted at 3hours after administration.  Also, if the authors cannot provide drug distribution at different time points, at least, the drug concentration in blood at different time points after administration of BPC157 should be provided to demonstrate the administration BPC157 didn't change the concentration of lithium in the blood. In addition, since the concentration of lithium in the blood is such high, perfusion is necessary when authors doing drug distribution studies.

Acknowledged. However, in our view, the early noted changes may be quite indicative for the subsequent course. As indicated, at the 20 min lithium-time, we already evidenced brain swelling and brain swelling counteraction (Figure 10), as well as muscle weakness (and muscle weakness counteraction) (Figure 19). Therefore, the 20 min time point seems to be well chosen, and thereby, the evidence about the similar concentration in the serum, as an argument, that the obtained beneficial effect is not related to the lithium concentration in the serum.

Of note, at that early time (i.e., 20 min lithium-time), lithium-overloaded BPC 157 rats with apparently attenuated lithium-course share the same high serum lithium concentration as the lithium-overloaded control-downhill injury course (for illustration, see Figure 10  (brain swelling, and BPC 157 effect), and Figure 19 (muscle weakness and BPC 157 effect), likely indicative for further either  downhill course (lithium-overloaded control rats)  or attenuated lithium-course (lithium-overloaded BPC 157 rats)). An alike situation appeared with the advanced lithium course (at 3 h lithium-time; lesser lithium concentration in the serum, higher in the brain [17]).

Finally, it seems to us that this reviewer’s discussion is a part related to our unintentional mistake (instead ppm, it should be ppb for the lithium serum concentration (Figure 20)), (however, the essential not difference in the serum concentration remains as a not disputed argument), and thereby, with apology for the all problems produced, this mistake is now fully corrected. Considering his/her point of the 3 h, providing that most of the experiments were conducted at 3 h after lithium administration, additional assessment was carried out. As mentioned, while these findings revealed the expected differences in lithium concentration related the time elapsed, the essential similar lithium concentration in serum of the all of the lithium overloaded rats remained, regardless very obvious difference in the lesions presentation. Therefore, it seems to us that we can confirm the previous conclusion.

In summary, we hope that the manuscript is now adequately improved, all technical points fully corrected, and finally, all comments made by the reviewer fully resolved.

Looking forward to hearing from you very soon

Sincerely

Predrag Sikiric, MD, PhD

Professor  

Round 3

Reviewer 2 Report

The article has been absolutely improved after modification. My question and concern have been almost solved.